# Long-Short Transformer: Efficient Transformers for Language and Vision

Chen Zhu[‡1*], Wei Ping[†2], Chaowei Xiao[†2,3], Mohammad Shoeybi[2], Tom Goldstein[1], Anima Anandkumar[2,4], and Bryan Catanzaro[2]

[1]University of Maryland, College Park
[2] NVIDIA    [3]Arizona State University  [4]California Institute of Technology
[‡]chenzhu@cs.umd.edu, [†]{wping, chaoweix}@nvidia.com

## Abstract

Transformers have achieved success in both language and vision domains. However, it is prohibitively expensive to scale them to long sequences such as long documents or high-resolution images, because self-attention mechanism has quadratic time and memory complexities with respect to the input sequence length. In this paper, we propose Long-Short Transformer (Transformer-LS), an efficient self-attention mechanism for modeling long sequences with linear complexity for both language and vision tasks. It aggregates a novel long-range attention with dynamic projection to model distant correlations and a short-term attention to capture fine-grained local correlations. We propose a dual normalization strategy to account for the scale mismatch between the two attention mechanisms. Transformer-LS can be applied to both autoregressive and bidirectional models without additional complexity. Our method outperforms the state-of-the-art models on multiple tasks in language and vision domains, including the Long Range Arena benchmark, autoregressive language modeling, and ImageNet classification. For instance, Transformer-LS achieves 0.97 test BPC on enwik8 using half the number of parameters than previous method, while being faster and is able to handle $3\times$ as long sequences compared to its full-attention version on the same hardware. On ImageNet, it can obtain the state-of-the-art results (e.g., a moderate size of 55.8M model solely trained on $224 \times 224$ ImageNet-1K can obtain Top-1 accuracy 84.1%), while being more scalable on high-resolution images. The source code and models are released at https://github.com/NVIDIA/transformer-ls.

## 1 Introduction

Transformer-based models [1] have achieved great success in the domains of natural language processing (NLP) [2, 3] and computer vision [4–6]. These models benefit from the self-attention module, which can capture both adjacent and long-range correlations between tokens while efficiently scaling on modern hardware. However, the time and memory consumed by self-attention scale quadratically with the input length, making it very expensive to process long sequences. Many language and vision tasks benefit from modeling long sequences. In NLP, document-level tasks require processing long articles [e.g., 7, 8], and the performance of language models often increases with sequence length [e.g., 9, 10]. In computer vision, many tasks involve high-resolution images, which are converted to long sequences of image patches before being processed with Transformer models [4, 6, 11]. As a result, it is crucial to design an efficient attention mechanism for long sequence modeling that generalizes well across different domains.

---

[*]Work done during an internship at NVIDIA.

35th Conference on Neural Information Processing Systems (NeurIPS 2021).

Various methods have been proposed to reduce the quadratic cost of full attention. However, an efficient attention mechanism that generalizes well in both language and vision domains is less explored. One family of methods is to sparsify the attention matrix with predefined patterns such as sliding windows [e.g., 12–15] and random sparse patterns [16]. These methods leverage strong inductive biases to improve both computational and model performance, but they limit the capacity of a self-attention layer because each specific token can only attend to a subset of tokens. Another family of methods leverages low-rank projections to form a low resolution representation of the input sequence, but the successful application of these methods has been limited to certain NLP tasks [e.g., 17–19]. Unlike sparse attention, this family of methods allows each token to attend to the entire input sequence. However, due to the loss of high-fidelity token-wise information, their performance sometimes is not as good as full attention or sparse attention on tasks that require fine-grained local information, including standard benchmarks in language [20] and vision [21].

Despite the rapid progress in efficient Transformers, some proposed architectures can only be applied to bidirectional models [e.g., 15, 16, 18]. Transformer-based autoregressive models have achieved great successes in language modeling [22], image synthesis [23], and text-to-image synthesis [24], which also involve long texts or high-resolution images. It is desirable to design an efficient transformer that can be applied to both autoregressive and bidirectional models.

In this work, we unify a local window attention and a novel long-range attention into a single efficient attention mechanism. We show that these two kinds of attention have complementary effects that together yield the state-of-the-art results on a range of tasks in language and vision, for both autoregressive and bidirectional models. Specifically, we make the following contributions:

- We propose Long-Short Transformer (Transformer-LS), an efficient Transformer that integrates a dynamic projection based attention to model long-range correlations, and a local window attention to capture fine-grained correlations. Transformer-LS can be applied to both autoregressive and bidirectional models with linear time and memory complexity.

- We compute a dynamic low-rank projection, which depends on the content of the input sequence. In contrast to previous low-rank projection methods, our dynamic projection method is more flexible and robust to semantic-preserving positional variations (e.g., insertion, paraphrasing). We demonstrate that it outperforms previous low-rank methods [17, 18] on Long Range Arena benchmark [20].

- We identify a scale mismatch problem between the embeddings from the long-range and short-term attentions, and design a simple but effective dual normalization strategy, termed *DualLN*, to account for the mismatch and enhance the effectiveness of the aggregation.

- We demonstrate that Long-Short Transformer, despite its low memory and runtime complexity, outperforms the state-of-the-art models on a set of tasks from Long Range Arena, and autoregressive language modeling on enwik8 and text8. In addition, the proposed efficient attention mechanism can be easily applied to the most recent vision transformer architectures [6, 11] and provides state-of-the-art results, while being more scalable to high-resolution images. We also investigate the robustness properties of the Transformer-LS on diverse ImageNet datasets.

## 2 Related Work

### 2.1 Efficient Transformers

In recent years, many methods have been introduced for dealing with the quadratic cost of full attention. In general, they can be categorized as follows: *i)* Sparse attention mechanism with predefined patterns (e.g., sliding window), including Sparse Transformer [12], Image Transformer [13], Axial Transformer [25] for modeling images, and Longformer [14], blockwise self-attention [26], ETC [15], Big Bird [16] for modeling language. *ii)* Low-rank projection attention, including Linformer [17], Nyströmformer [18], Synthesizer [19]. For example, Linformer uses linear layers to project the original high resolution keys ($K$) and values ($V$) with length $n$ to low resolution with size $r$ ($r \ll n$) and allows all query tokens ($Q$) to attend these compressed representations. *iii)* Memory-based mechanisms like Compressive Transformer [10] and Set Transformer [27], which use extra memories for caching global long-range information for use in computing attention between distant tokens. *iv)* Kernel-based approximation of the attention matrix, including Performer [28], Linear

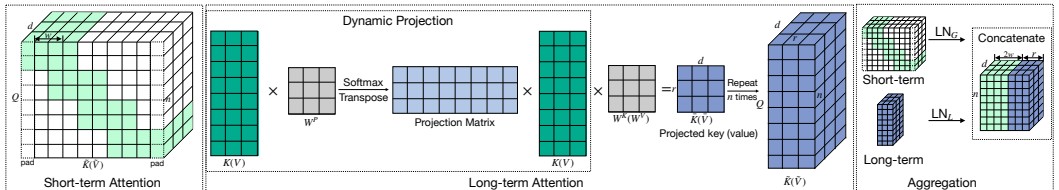

Figure 1: Long-short term attention of a single attention head. Here, the sequence length $n = 8$, hidden dimension $d = 3$, local window segment size $w = 2$, and rank of dynamic projection $r = 3$. Within the figure, $K(V)$ denotes key $K$ or value $V$. In the left figure, we virtually replicate $K$ or $V \in \mathbb{R}^{n \times d}$ into $n$ rows, and highlight the keys and values within the attention span (denoted as $\tilde{K}(\tilde{V})$) of all $n$ queries $Q$ for the short-term attention. In the middle figure, all queries attend to the same projected keys $\bar{K}$ and values $\bar{V}$ within the long-term attention. In the right figure, $\tilde{K}(\tilde{V})$ and $\bar{K}(\bar{V})$ are first normalized with two sets of LayerNorms, and the queries attend to normalized $\tilde{K}(\tilde{V})$ and $\bar{K}(\bar{V})$ within their attention span simultaneously.

Transformer [29], and Random Feature Attention [30]. *vi)* Similarity and clustering based methods, including Reformer [31], Routing Transformer [32], and Sinkhorn Transformer [33].

Our method seamlessly integrates both low-rank projection and local window attentions, to leverage their strengths for modeling long-range and short-term correlations. In particular, our long-range attention uses a dynamic low-rank projection to encode the input sequence, and outperforms the previous low-rank projection method used by the Linformer [17]. In the similar vein, a few other methods also try to combine the strengths of different methods. For example, Longformer [14] and ETC [15] augment local window attention with task motivated global tokens. Such global tokens may not be applicable for some tasks (e.g., autoregressive modelling). BigBird [16] further combines local window and global token attention with random sparse attention. It is not applicable in autoregressive tasks because the global token and random sparse pattern are introduced. To compress the model footprint on edge devices, Lite Transformer [34] combines convolution and self-attention, but it still has quadratic complexity for long sequences.

## 2.2 Vision Transformers

Vision Transformer (ViT) [4] splits images as small patches and treats the patches as the input word tokens. It uses a standard transformer for image classification and has shown to outperform convolutional neural networks (e.g., ResNet [35]) with sufficient training data. DeiT [36] has applied the teacher-student strategy to alleviate the data efficiency problem of ViT and has shown strong comparable performance using only the standard ImageNet dataset [37]. Instead of applying transformer at a single low resolution of patches (e.g., $16 \times 16$ patches), very recent works, including Pyramid Vision Transformer (PVT) [5], Swin-Transformer [38], T2T-ViT [39], Vision Longformer (ViL) [11] and Convolutional Vision Transformer (CvT) [6], stack a pyramid of ViTs to form a multi-scale architecture and model long sequences of image patches at much higher resolution (e.g., $56 \times 56 = 3136$ patches for images with $224 \times 224$ pixels). Most of these methods have quadratic complexity of self-attention with respect to the input image size.

To reduce the complexity, Swin-Transformer [38] achieves linear complexity by limiting the computation of self-attention only within each local window. HaloNet [40] applies local attention on blocked images and only has quadratic complexity with respect to the size of the block. Perceiver [41] uses cross-attention between data and latent arrays to replace the self-attention on data to remove the quadratic complexity bottleneck. Vision Longformer (ViL) [11], another concurrent work, achieves linear complexity by adapting Longformer [14] to Vision. ViL augments local window attention with task-specific global tokens, but the global tokens are not applicable for decoding task (e.g., image synthesis [23, 24]). In contrast, our method reduces the quadratic cost to linear cost by combining local window attention with global dynamic projection attention, which can be applied to both encoding and decoding tasks.

## 3 Long-Short Transformer

Transformer-LS approximates the full attention by aggregating long-range and short-term attentions, while maintaining its ability to capture correlations between all input tokens. In this section, we first

introduce the preliminaries of multi-head attention in Transformer. Then, we present the short-term attention via sliding window, and long-range attention via dynamic projection, respectively. After that, we propose the aggregating method and dual normalization (DualLN) strategy. See Figure 1 for an illustration of our long-short term attention.

## 3.1 Preliminaries and Notations

Multi-head attention is a core component of the Transformer [1], which computes contextual representations for each token by attending to the whole input sequence at different representation subspaces. It is defined as

$$\text{MultiHead}(Q, K, V) = \text{Concat}(H_1, H_2, ..., H_h)W^O, \tag{1}$$

where $Q, K, V \in \mathbb{R}^{n \times d}$ are the query, key and value embeddings, $W^O \in \mathbb{R}^{d \times d}$ is the projection matrix for output, the $i$-th head $H_i \in \mathbb{R}^{n \times d_k}$ is the scaled dot-product attention, and $d_k = d/h$ is the embedding dimension of each head,

$$H_i = \text{Attention}(QW_i^Q, KW_i^K, VW_i^V) = \text{softmax}\left[\frac{QW_i^Q \left(KW_i^K\right)^\mathsf{T}}{\sqrt{d_k}}\right] VW_i^V = A_i VW_i^V, \tag{2}$$

where $W_i^Q, W_i^K, W_i^V \in \mathbb{R}^{d \times d_k}$ are learned projection matrices, and $A_i \in \mathbb{R}^{n \times n}$ denotes the full attention matrix for each attention head. The complexity of computing and storing $A_i$ is $O(n^2)$, which can be prohibitive when $n$ is large. For simplicity, our discussion below is based on the case of 1D input sequences. It is straightforward to extend to the 2D image data given a predetermined order.

## 3.2 Short-term Attention via Segment-wise Sliding Window

We use the simple yet effective sliding window attention to capture fine-grained local correlations, where each query attends to nearby tokens within a fixed-size neighborhood. Similar techniques have also been adopted in [14, 16, 11]. Specifically, we divide the input sequence into disjoint segments with length $w$ for efficiency reason. All tokens within a segment attend to all tokens within its home segment, as well as $w/2$ consecutive tokens on the left and right side of its home segment (zero-padding when necessary), resulting in an attention span over a total of $2w$ key-value pairs. See Figure 5 in Appendix for an illustration. For each query $Q_t$ at the position $t$ within the $i$-th head, we denote the $2w$ key-value pairs within its window as $\tilde{K}_t, \tilde{V}_t \in \mathbb{R}^{2w \times d}$. For implementation with PyTorch, this segment-wise sliding window attention is faster than the per-token sliding window attention where each token attends to itself and $w$ tokens to its left and right, and its memory consumption scales linearly with sequence length; see [14] and our Figure 3 for more details.

The sliding window attention can be augmented to capture long-range correlations in part, by introducing different dilations to different heads of sliding window attention [14]. However, the dilation configurations for different heads need further tuning and an efficient implementation of multi-head attention with different dilations is non-trivial. A more efficient alternative is to augment sliding window attention with random sparse attention [16], but this does not guarantee that the long-range correlations are captured in each layer as in full attention. In the following section, we propose our long-range attention to address this issue.

## 3.3 Long-range Attention via Dynamic Projections

Previous works have shown that the self-attention matrix can be well approximated by the product of low-rank matrices [17]. By replacing the full attention with the product of low-rank matrices [42, 19, 18, 43, 28], each query is able to attend to all tokens. Linformer [17] is one of the most representative models in this category. It learns a fixed projection matrix to reduce the length of the keys and values, but the fixed projection is inflexible to semantic-preserving positional variations.

Starting from these observations, we parameterize the dynamic low-rank projection at $i$-th head as $P_i = f(K) \in \mathbb{R}^{n \times r}$, where $r \ll n$ is the low rank size and $P_i$ depends on all the keys $K \in \mathbb{R}^{n \times d}$ of input sequence. It projects the $(n \times d_k)$-dimensional key embeddings $KW_i^K$ and value embeddings $VW_i^V$ into shorter, $(r \times d_k)$-dimensional key $\bar{K}_i$ and value $\bar{V}_i$ embeddings. Unlike Linformer [17], the low-rank projection matrix is dynamic, which depends on the input sequence and is intended to be more flexible and robust to, e.g., insertion, deletion, paraphrasing, and other operations that change sequence length. See Table 2 for examples. Note that, the query embeddings $QW_i^Q \in \mathbb{R}^{n \times d_k}$ are kept

at the same length, and we let each query attend to $\bar{K}_i$ and $\bar{V}_i$. In this way, the full $(n \times n)$ attention matrix can be decomposed into the product of two matrices with $r$ columns or rows. Specifically, we define the dynamic projection matrix $P_i \in \mathbb{R}^{n \times r}$ and the key-value embeddings $\bar{K}_i, \bar{V}_i \in \mathbb{R}^{r \times d_k}$ of low-rank attention as

$$P_i = \text{softmax}(KW_i^P), \ \bar{K}_i = P_i^\intercal KW_i^K, \bar{V}_i = P_i^\intercal VW_i^V, \tag{3}$$

where $W_i^P \in \mathbb{R}^{d \times r}$ are learnable parameters,[2] and the softmax normalizes the projection weights on the first dimension over all $n$ tokens, which stabilizes training in our experiments. Note that $K = V$ in all the experiments we have considered, so $P_i$ remains the same if it depends on $V$. The computational complexity of Eq. 3 is $O(rn)$.

To see how the full attention is replaced by the product of low-rank matrices, we compute each head $H_i \in \mathbb{R}^{n \times d_k}$ of long-range attention as,

$$\bar{H}_i = \underbrace{\text{softmax}\left[\frac{QW_i^Q \bar{K}_i^\intercal}{\sqrt{d_k}}\right]}_{\bar{A}_i} \bar{V}_i = \bar{A}_i \big(P_i^\intercal VW_i^V\big), \tag{4}$$

so the full attention is now replaced with the implicit product of two low-rank matrices $\bar{A}_i \in \mathbb{R}^{n \times r}$ and $P_i^\intercal \in \mathbb{R}^{r \times n}$, and the computational complexity is reduced to $O(rn)$. Note the effective attention weights of a query on all tokens still sum to 1. Our global attention allows each query to attend to all token embeddings within the same self-attention layer. In contrast, the sparse attention mechanisms [14, 16] need stack multiple layers to build such correlations.

**Application to Autoregressive Models:** In autoregressive models, each token can only attend to the previous tokens, so the long-range attention should have a different range for different tokens. A straightforward way to implement our global attention is to update $\bar{K}_i, \bar{V}_i$ for each query recurrently, but this requires re-computing the projection in Eq. (3) for every token due to the nonlinearity of softmax, which results in $O(rn^2)$ computational complexity. To preserve the linear complexity, for autoregressive models, we first divide the input sequence into equal-length segments with length $l$, and apply our dynamic projection to extract $\bar{K}_i, \bar{V}_i$ from each segment. Each token can only attend to $\bar{K}_i, \bar{V}_i$ of segments that do not contain its future tokens. Formally, let $Q_t$ be the query at position $t$, $K_{(l-1)s:ls}, V_{(l-1)s:ls}$ be the key-value pairs from the $s$-th segment, and $s_t = \lfloor t/l \rfloor$. For autoregressive models, we compute the long-range attention of $Q_t$ by attending to $\bar{K}_{i,t}, \bar{V}_{i,t}$, defined as

$$\bar{K}_{i,t} = [P_{i,1}^\intercal K_{1:l}; ...; P_{i,s_t}^\intercal K_{(l-1)s_t:ls_t}]W_i^K, \bar{V}_{i,t} = [P_{i,1}^\intercal V_{1:l}; ...; P_{i,s_t}^\intercal V_{(l-1)s_t:ls_t}]W_i^V. \tag{5}$$

In this way, the dynamic low-rank projection is applied to each segment only once in parallel, preserving the linear complexity and the high training speed. By comparison, Random Feature Attention [30] is slow at training due to the requirement for recurrence.

### 3.4 Aggregating Long-range and Short-term Attentions

To aggregate the local and long-range attentions, instead of adopting different attention mechanisms for different heads [12, 14, 34], we let each query at $i$-th head attend to the union of keys and values from the local window and global low-rank projections, thus it can learn to select important information from either of them. We find this aggregation strategy works better than separating the heads in our initial trials with the autoregressive language models. Specifically, for the $i$-th head, we denote the global low-rank projected keys and values as $\bar{K}_i, \bar{V}_i \in \mathbb{R}^{r \times d_k}$, and the local keys and values as $\tilde{K}_t, \tilde{V}_t \in \mathbb{R}^{2w \times d}$ within the local window of position $t$ for the query $Q_t$. Then the $i$-th attention $H_{i,t}$ at position $t$ is

$$H_{i,t} = \text{softmax}\left[\frac{Q_t W_i^Q \left[\tilde{K}_t W_i^K; \bar{K}_i\right]^\intercal}{\sqrt{d_k}}\right][\tilde{V}_t W_i^V; \bar{V}_i]. \tag{6}$$

where $[\cdot\,;\cdot]$ denotes concatenating the matrices along the first dimension. Furthermore, we find a scale mismatch between the initial norms of $\tilde{K}_t W_i^K$ and $\bar{K}_i$, which biases the attention to the local window at initialization for both language and vision tasks. We introduce a normalization strategy (DualLN) to align the norms and improve the effectiveness of the aggregation in the following.

---

[2]For the CvT-based vision transformer model, we replace $W_i^P$ with a depth-wise separable convolution, just as its query, key and value projections.

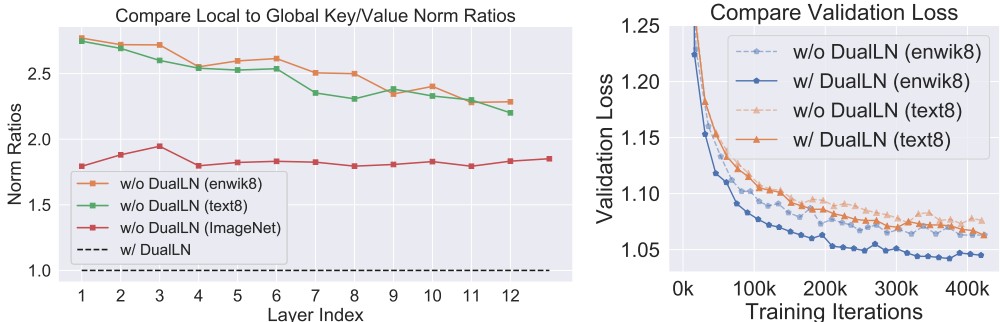

Figure 2: Left: Ratios of the average $\ell_2$ norms of the local window to global low-rank key/value embeddings at initialization. Without DualLN, the sparse and low-rank embeddings have a magnitude mismatch. With DualLN, the ratios will be $1.0$ at every layer, which will facilitate optimization. Right: The validation loss of Transformer-LS with and without DualLN on enwik8 and text8.

**DualLN:** For Transformers with Layer Normalization (LN) (see [44] for an illustration), the $K_i, V_i$ embeddings are the outputs of LN layers, so they have zero mean and unit variance at initialization. The $\ell_2$ norm of vectors with zero-mean entries is proportional to their variance in expectation. We note a weighted average will reduce the variance and therefore the norm of such zero-mean vectors. As a result, the embedding vectors from low-rank attention in the weighted average $\bar{K}_i, \bar{V}_i$ of Eq. (3) will have smaller norms than the regular key and value embeddings from sliding window attention (see Figure 2 Left for an illustration). This scale mismatch causes two side effects. First, the inner product $Q_t W_i^Q \bar{K}_i^\mathsf{T}$ from local-rank component tends to have smaller magnitude than the local window one, thus the attention scores on long-range attention is systematically smaller. Second, the key-value pairs $\bar{K}_i, \bar{V}_i$ for the low-rank attention will naturally have less impact on the direction of $H_i$ even when low-rank and local window are assigned with same attention scores, since $\bar{V}_i$ has smaller norms. Both effects lead to small gradients on the low-rank components and hinders the model from learning to effectively use the long-range correlations.

To avoid such issues, we add two sets of Layer Normalizations after the key and value projections for the local window and global low-rank attentions, so that their scales are aligned at initialization, but the network can still learn to re-weight the norms after training. Specifically, the aggregated attention is now computed as

$$H_{i,t} = \text{softmax}\left[\frac{Q_t W_i^Q \left[\text{LN}_L(\tilde{K}_t W_i^K); \text{LN}_G(\bar{K}_i)\right]^\mathsf{T}}{\sqrt{d_k}}\right][\text{LN}_L(\tilde{V}_t W_i^V); \text{LN}_G(\bar{V}_i)], \qquad (7)$$

where $\text{LN}_L(\cdot), \text{LN}_G(\cdot)$ denote the Layer Normalizations for the local and global attentions respectively. In practice, to maintain the consistency between the local attention and dynamic projection, we use $\text{LN}_L(K), \text{LN}_L(V)$ instead of $K, V$ to compute $\bar{K}_i, \bar{V}_i$ in Eq. 3. As illustrated in Figure 2 Right, the Transformer-LS models trained with DualLN has consistently lower validation loss than the models without DualLN.

## 4 Experiments

In this section, we demonstrate the effectiveness and efficiency of our method in both language and vision domains. We use PyTorch for implementation and count the FLOPs using fvcore [45].

### 4.1 Bidirectional Modeling on Long Range Arena and IMDb

To evaluate Long-Short Transformer as a bidirectional encoder for long text, we train our models on the three NLP tasks, **ListOps**, **Text**, and **Retrieval**, from the recently proposed Long Range Arena (LRA) benchmark [20], following the setting of Peng et al. [30] and Tay et al. [46]. For fair comparisons, we use the PyTorch implementation and the same data preprocessing/split, training hyperparameters and model size from [18], except for **Retrieval** where we accidentally used more warmup steps and improved the results for all models. See Appendix B for more details. The results

Table 1: Accuracy (%) and FLOPs (G) on Long Range Arena (LRA), with the model configs annotated (see Table 7 for more). All results are averages of 4 runs with different random seeds.

| Task | ListOps | | Text | | Retrieval | | Average |
|---|---|---|---|---|---|---|---|
| (mean ± std.) of sequence length | (888 ± 339) | | (1296 ± 893) | | (3987 ± 560) | | |
| Model | Acc. | FLOPs | Acc. | FLOPs | Acc. | FLOPs | Acc. |
| Full Attention [1] | 37.13 | 1.21 | 65.35 | 4.57 | **82.30** | 9.14 | 61.59 |
| Reformer [31] (2) | 36.44 | 0.27 | 64.88 | 0.58 | 78.64 | 1.15 | 59.99 |
| Linformer [17] ($k$=256) | 37.38 | 0.41 | 56.12 | 0.81 | 79.37 | 1.62 | 57.62 |
| Performer [28] ($r = 256$) | 32.78 | 0.41 | 65.21 | 0.82 | 81.70 | 1.63 | 59.90 |
| Nyströmformer [18] ($l = 128$) | 37.34 | 0.61 | 65.75 | 1.02 | 81.29 | 2.03 | 61.46 |
| Transformer-LS ($w, r = 8, 32$) | **37.50** | 0.20 | **66.01** | 0.40 | 81.79 | 0.80 | **61.77** |
| Dynamic Projection (best) | 37.79 | 0.15 | 66.28 | 0.69 | 81.86 | 2.17 | 61.98 |
| Transformer-LS (best) | **38.36** | 0.16 | **68.40** | 0.29 | **81.95** | 2.17 | **62.90** |

Table 2: Comparing the robustness of the models under test-time insertions and deletions. DP refers to long-range attention via Dynamic Projection, and Win. refers to sliding window attention.

| Task | Text | | | Retrieval | | |
|---|---|---|---|---|---|---|
| Test Perturb | None | Insertion | Deletion | None | Insertion | Deletion |
| Linformer | 56.12 | 55.94 | 54.91 | 79.37 | 53.66 | 51.75 |
| DP | 66.28 | 63.16 | 58.95 | 81.86 | **70.01** | **64.98** |
| Linformer + Win. | 59.63 | 56.69 | 56.29 | 79.68 | 52.83 | 52.13 |
| DP + Win. (ours) | **68.40** | **66.34** | **62.62** | **81.95** | 69.93 | 64.19 |

Table 3: Comparing the results of pretrained language models fine-tuned on IMDb.

| Model | RoBERTa-base | RoBERTa-large | Longformer-base | LS-base | LS-large |
|---|---|---|---|---|---|
| Accuracy | 95.3 | 96.5 | 95.7 | 96.0 | **96.8** |

on these three tasks are given in Table 1. Results of the other two image-based tasks of LRA, as well as models implemented in JAX, are given in Appendix C and C.2.

In addition, we follow the pretraining procedure of Longformer [14] to pretrain our models based on RoBERTa-base and RoBERTa-large [47], and fine-tune it on the IMDb sentiment classification dataset. The results are given in Table 3.

**Results.** From Table 3, our base model outperforms Longformer-base, and our large model achieves improvements over RoBERTa-large, demonstrating the benefits of learning to model long sequences. Comparisons with models on LRA are given in Table 1. Transformer-LS (best) with the best configurations of $w, r$ for each task are given in Table 7 in Appendix B. We also report the results of using fixed hyperparameter $w = 8, r = 32$ on all tasks. Overall, our Transformer-LS (best) is significantly better than other efficient Transformers, and the model with $w, r = 8, 32$ performs favorably while using only about 50% to 70% computation compared to other efficient Transformers on all three tasks. The advantage of aggregating local and long-range attentions is the most significant on **ListOps**, which requires the model to understand the tree structures involving both long-term and short-term relations. On **Retrieval**, where document-level encoding capability is tested, we find our global attention more effective than window attention. The test accuracy of using only dynamic projection is about 10% higher than Linformer on **Text** (i.e., 66.28 vs. 56.12), which has the highest variance in sequence length (i.e. standard deviation 893). This demonstrates the improved flexibility of dynamic projection at learning representations for data with high variance in sequence length, compared to the learned but fixed projection of Linformer. Similarly, Linformer, Nyströmformer and our model outperform full attention on **ListOps**, indicating they may have better inductive bias, and efficient Transformers can have better efficacy beyond efficiency.

**Robustness of Dynamic Projection.** In Table 2, we compare the robustness of Linformer and the proposed Dynamic Projection (DP) against insertion and deletion on Text and Retrieval tasks of LRA. We train the models on the original, clean training sets and only perturb their test sets. For insertion, we insert 10 random punctuations at 10 random locations of each test sample. For deletion, we delete

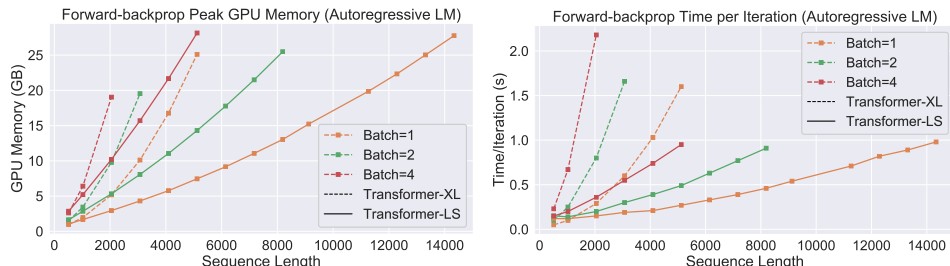

Figure 3: Running time and memory consumption of Transformer-XL (full attention) and our Transformer-LS on Char-LM. We increase the sequence length until we use up the 32GB of memory on a V100 GPU. Transformer-LS is the same smaller model in Table 4. We use dashed lines to represent the full attention Transformer and solid lines to represent our model. We use different colors to represent different batch sizes.

Table 4: BPC (↓) of smaller models on enwik8 and text8 (left), and larger models on enwik8 (right).

| Method | #Param | text8 | | enwik8 | |
| | | Dev | Test | Dev | Test |
|---|---|---|---|---|---|
| T12 [49] | 44M | - | 1.18 | - | 1.11 |
| Transformer-XL [9] | 41M | - | - | - | 1.06 |
| Reformer [31] | - | - | - | - | 1.05 |
| Adaptive [50] | 38M | 1.05 | 1.11 | 1.04 | 1.02 |
| BP-Transformer [51] | 38M | - | 1.11 | - | 1.02 |
| Longformer [20] | 41M | 1.04 | 1.10 | 1.02 | 1.00 |
| Transformer-LS | 44M | 1.03 | **1.09** | 1.01 | **0.99** |

| Method | #Param | Test BPC |
|---|---|---|
| Transformer-XL [9] | 88M | 1.03 |
| Transformer-XL [9] | 277M | 0.99 |
| Routing [32] | 223M | 0.99 |
| Longformer [14] | 102M | 0.99 |
| Sparse [12] | 95M | 0.99 |
| Adaptive [50] | 209M | 0.98 |
| Compressive [10] | 227M | **0.97** |
| Transformer-LS | 110M | **0.97** |

all punctuations from the test samples. Both transforms are label-preserving in most cases. By design, dynamic projection is more robust against location changes.

## 4.2 Autoregressive Language Modeling

We compare our method with other efficient transformers on the character-level language modeling where each input token is a character.

**Setup.** We train and evaluate our model on enwik8 and text8, each with 100M characters and are divided into 90M, 5M, 5M for train, dev, test, following [48]. Our smaller 12-layer and larger 30-layer models are Pre-LN Transformers with the same width and depth as Longformer [20], except that we add relative position encoding to the projected segments in each layer. We adopt the cache mechanism of Transformer-XL [9], setting the cache size to be the same as the input sequence length. We follow similar training schedule as Longformer, and train our model in 3 phases with increasing sequence lengths. The input sequence lengths are 2048, 4096 and 8192 respectively for the 3 phases. By comparison, Longformer trains their model in 5 phases on GPUs with 48GB memory (The maximal of ours is 32GB) where the sequence length is 23,040 in the last phase. The window size of Longformer increases with depth and its average window size is 4352 in phase 5, while our effective number of attended tokens is 1280 on average in the last phase. Each experiment takes around 8 days to finish on 8 V100 GPUs. Detailed hyperparameters are shown in Appendix D. For testing, same as Longformer, we split the dataset into overlapping sequences of length 32K at a step size of 512, and evaluate the BPCs for predicting the next 512 tokens given the previous 32K characters.

**Results** Table 4 shows comparisons on text8 and enwik8. Our method has achieved state-of-the-art results. On text8, we achieve a test BPC of 1.09 with the smaller model. On enwik8, our smaller model achieves a test BPC of 0.99, and outperforms the state-of-the-art models with comparable number of parameters. Our larger model obtains a test BPC of 0.97, on par with the Compressive Transformer with 2× parameters. Our results are consistently better than Longformer which is trained on longer sequences with 5 stages and 48 GPU memory. In Figure 3, we show our model is much more memory and computational efficient than full attention.

Table 5: Test accuracies on ImageNet, ImageNet Real [52], and ImageNet V2 [53] of models trained on ImageNet-1K. **Grey-colored rows are our results**. CvT*-LS denotes our long-short term attention based on the non-official CvT implementation. ViL models with LS suffixes are our long-short term attention based on the official ViL implementation with relative positional bias. We also provide the latency of models tested using batch size 32 on the same V100 GPU. Our improvements over ViL is mainly from a better implementation of the short-term attention.

| Model | #Param (M) | Image Size | FLOPs (G) | ImageNet top-1 (%) | Real top-1 (%) | V2 top-1 (%) | Latency (s) |
|---|---|---|---|---|---|---|---|
| ResNet-50 | 25 | $224^2$ | 4.1 | 76.2 | 82.5 | 63.3 | - |
| ResNet-101 | 45 | $224^2$ | 7.9 | 77.4 | 83.7 | 65.7 | - |
| ResNet-152 | 60 | $224^2$ | 11 | 78.3 | 84.1 | 67.0 | - |
| DeiT-S [36] | 22 | $224^2$ | 4.6 | 79.8 | 85.7 | 68.5 | - |
| DeiT-B [36] | 86 | $224^2$ | 17.6 | 81.8 | 86.7 | 70.9 | - |
| PVT-Medium [5] | 44 | $224^2$ | 6.7 | 81.2 | - | - | - |
| PVT-Large [5] | 61 | $224^2$ | 9.8 | 81.7 | - | - | - |
| Swin-S [38] | 50 | $224^2$ | 8.7 | 83.2 | - | - | - |
| Swin-B [38] | 88 | $224^2$ | 15.4 | 83.5 | - | - | 0.115 |
| PVTv2-B4 [54] | 62.6 | $224^2$ | 10.1 | 83.6 | - | - | - |
| PVTv2-B5 [54] | 82.0 | $224^2$ | 11.8 | 83.8 | - | - | - |
| ViT-B/16 [4] | 86 | $384^2$ | 55.5 | 77.9 | - | - | - |
| ViT-L/16 [4] | 307 | $384^2$ | 191.1 | 76.5 | - | - | - |
| DeiT-B [36] | 86 | $384^2$ | 55.5 | 83.1 | - | - | - |
| Swin-B [38] | 88 | $384^2$ | 47.1 | **84.5** | - | - | 0.378 |
| CvT-13 [6] | 20 | $224^2$ | 6.7 | 81.6 | 86.7 | 70.4 | 0.122 |
| CvT-21 [6] | 32 | $224^2$ | 10.1 | 82.5 | 87.2 | 71.3 | 0.165 |
| CvT*-LS-13 | 20.3 | $224^2$ | 4.9 | 81.9 | 87.0 | 70.5 | 0.083 |
| CvT*-LS-17 | 23.7 | $224^2$ | 9.8 | 82.5 | 87.2 | 71.6 | - |
| CvT*-LS-21 | 32.1 | $224^2$ | 7.9 | 82.7 | 87.5 | 71.9 | 0.122 |
| CvT*-LS-21S | 30.1 | $224^2$ | 11.3 | **82.9** | 87.4 | 71.7 | - |
| CvT-13 [6] | 20 | $384^2$ | 31.9 | 83.0 | 87.9 | 71.9 | - |
| CvT-21 [6] | 32 | $384^2$ | 45.0 | 83.3 | 87.7 | 71.9 | - |
| CvT*-LS-21 | 32.1 | $384^2$ | 23.9 | 83.2 | 88.0 | 72.5 | - |
| CvT*-LS-21 | 32.1 | $448^2$ | 34.2 | **83.6** | 88.2 | 72.9 | - |
| ViL-Small [14] | 24.6 | $224^2$ | 4.9 | 82.4 | - | - | - |
| ViL-Medium [14] | 39.7 | $224^2$ | 8.7 | 83.5 | - | - | 0.106 |
| ViL-Base [14] | 55.7 | $224^2$ | 13.4 | 83.7 | - | - | 0.164 |
| ViL-LS-Medium | 39.8 | $224^2$ | 8.7 | 83.8 | - | - | 0.075 |
| ViL-LS-Base | 55.8 | $224^2$ | 13.4 | **84.1** | - | - | 0.113 |
| ViL-LS-Medium | 39.9 | $384^2$ | 28.7 | **84.4** | - | - | 0.271 |

## 4.3 ImageNet Classification

We train and evaluate the models on ImageNet-1K with 1.3M images and 1K classes. We use CvT [6] and ViL [11], state-of-the art vision transformer architectures, as the backbones and replace their attention mechanisms with our long-short term attention, denoted as CvT*-LS and ViL-size-LS in Table 5. CvT uses overlapping convolutions to extract dense patch embeddings from the input images and feature maps, resulting in a long sequence length in the early stages (e.g., $56 \times 56 = 3136$ patches for images with $224^2$ pixels). For ViL, our sliding window uses the same group size $w$, but each token attends to at most $2w \times 2w$ (rounding when necessary) tokens inside the window, instead of $3w \times 3w$ as ViL, which allows adding our dynamic projection without increasing the FLOPs. We set $r = 8$ for the dynamic projections for both ViL-LS-Medium and ViL-LS-Base. Note that, our efficient attention mechanism does not depend on the particular architecture, and it can be applied to other vision transformers [e.g., 4, 36, 5]. Please refer to Appendix E for more details.

**Classification Results.** The results are shown in the Table 5, where we also list test accuracies on ImageNet Real and ImageNet V2. Except for CvT, we compare with the original ViT [4] and the enhanced DeiT [36], PVT [5] that also uses multi-scale stragey, ViL [11] that uses window attention and global tokens to improve the efficiency. Training at high-resolution usually improves the test

Table 6: Robustness evaluation on various ImageNet datasets. Top-1/Acc.: Top-1 accuracy. mCE: Mean Corruption Error. Mixed-same/Mixed-rand: accuracies on MIXED-SAME/MIXED-RAND subsets.

| Model | Params | ImageNet | IN-C [56] | IN-A [57] | IN-R [58] | ImageNet-9 [59] | |
|---|---|---|---|---|---|---|---|
| | (M) | Top-1 | mCE ($\downarrow$) | Acc. | Acc. | Mixed-same | Mixed-rand |
| ResNet-50 [35] | 25.6 | 76.2 | 78.9 | 6.2 | 35.3 | 87.1 | 81.6 |
| DeiT-S [36] | 22.1 | 79.8 | 57.1 | 19.0 | 41.9 | 89.1 | 84.2 |
| CvT-13 | 20 | 81.6 | 59.6 | 25.4 | 42.9 | 90.5 | 85.7 |
| CvT-21 | 32 | 82.5 | 56.2 | **31.1** | 42.6 | 90.5 | 85.0 |
| CvT*-LS-13 | 20.3 | 81.9 | 58.7 | 27.0 | 42.6 | 90.7 | 85.6 |
| CvT*-LS-21 | 32.1 | **82.7** | **55.2** | 29.3 | **45.0** | **91.5** | **85.8** |

accuracy of vision transformer. With our long-short term attention, we can easily scale the training to higher resolution, and the performance of CvT*-LS and ViL-LS also improves. Our best model with CvT (CvT*-LS-21 at $448^2$) achieves 0.3% higher accuracy than the best reported result of CvT while using the same amount of parameters and 76% of its FLOPs. In CvT architecture, the spatial dimension of feature maps in earlier stages are large, representing more fine-grained details of the image. Similar to training with high-resolution images, the model should also benefit from denser feature maps. With our efficient long-short term attention, we can better utilize these fine-grained feature maps with less concerns about the computational budget. In this way, our CvT*-LS-17 achieves better result than CvT-21 at resolution 224 using fewer parameters and FLOPs, and our CvT*-LS-21S model further improves our CvT*-LS-21 model.

Our ViL-LS-Medium and ViL-LS-Base with long-short term attention improve the accuracies of ViL-Medium and ViL-Base from 83.5 and 83.7 to 83.8 and 84.1 respectively, without an increase in FLOPs. When increasing the resolution for training ViL-LS-Medium from $224^2$ to $384^2$, the FLOPs increased (approximately) linearly and the accuracy improved by 0.6%, showing our method still benefits greatly from increased resolution while maintaining the linear complexity in practice.

**Short-term Attention Suppresses Oversmoothing.** By restricting tokens from different segments to attend to different windows, our short-term sparse local attention encourages diversity of the feature representations and helps to alleviate the over-smoothing problem [55] (where all queries extract similar information in deeper layers and the attention mechanism is less important), thus can fully utilize the depth of the network. As in [55], we provide the cosine similarity of patch embeddings of our CvT*-LS-13 and re-implemented CvT-13 (81.1 accuracy) in Figure 6 within Appendix. This is one of the reasons why our efficient attention mechanism can get even better results than the full attention CvT model in the same setting.

**Robustness evaluation on Diverse ImageNet Datasets.** As vision models have been widely used in safety-critical applications (e.g. autonomous driving), their robustness is vital. In addition to out-of-distribution robustness (ImageNet-Real and Imageet-v2), we further investigate the robustness of our vision transformer against common corruption (ImageNet-C), semantic shifts (ImageNet-R), Background dependence (ImageNet-9) and natural adversarial examples (ImageNet-A). We compare our methods with standard classification methods, including CNN-based model (ResNet [35]) and Transformer-based models (DeiT [36]) with similar numbers of parameters. As shown in Table 6, we observe that our method significantly outperforms the CNN-based method (ResNet-50). Compared to DeiT, our models also achieve favorable improvements. These results indicate that the design of different attention mechanisms plays an important role for model robustness, which sheds new light on the design of robust vision transformers. More details and results can be found in Appendix E.

## 5   Conclusion

In this paper, we introduced Long-Short Transformer, an efficient transformer for long sequence modeling for both language and vision domain, including both bidirectional and autoregressive models. We design a novel global attention mechanism with linear computational and memory complexity in sequence length based on a dynamic projection. We identify the scale mismatch issue and propose the DualLN technique to eliminate the mismatch at initialization and more effectively aggregate the local and global attentions. We demonstrate that our method obtains the state-of-the-art results on the Long Range Arena, char-level language modeling and ImageNet classification. We look forward to extending our methods to more domains, including document QA, object detection and semantic segmentation on high-resolution images.

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
