# Long-Short Transformer: Efficient Transformers for Language and Vision (Appendix)

## A  Details of Norm Comparisons

As we have shown in Figure 2, the norms of the key-value embeddings from the long-term and short-term attentions, $\bar{K}, \bar{V}$ and $\tilde{K}, \tilde{V}$, are different at initialization, and the norms of $\tilde{K}, \tilde{V}$ is always larger than $\bar{K}, \bar{V}$ on different networks and datasets we have evaluated. Here, we give an explanation.

Intuitively, at initialization, following similar assumptions as [60, 61], the entries of $K, V$ should have zero mean. Since each entry of $\bar{K}, \bar{V}$ is a weighted mean of $K, V$, they have smaller variance unless one of the weights is 1. Given that $\bar{K}, \bar{V}$ are also zero-mean, the norm of their embedding vectors (their rows), which is proportional to the variance, is smaller. For the key-value embeddings from short-term attention, $\tilde{K}, \tilde{V}$ are just a subset of $K, V$, so their embedding vectors should have the same norm as rows of $K, V$ in expectation. Therefore, the norms of embedding vectors from $\bar{K}, \bar{V}$ will be smaller than those from $\tilde{K}, \tilde{V}$ in expectation.

## B  Details for Experiments on Long Range Arena

**The tasks.**   We compare our method with the following three tasks:

- **ListOps**. ListOps [62] is designed to measure the parsing ability of models through hierarchically structured data. We follow the setting in [20] in which each instance contains 500-2000 tokens.

- **Text**. This is a binary sentiment classification task of predicting whether a movie review from IMDb is positive or negative [63]. Making correct predictions requires a model to reason with compositional unsegmented char-level long sequences with a maximum length of 4k.

- **Retrieval**. This task is based on the ACL Anthology Network dataset [64]. The model needs to classify whether there is a common citation between a pair of papers, which evaluates the model's ability to encode long sequences for similarity-based matching. The max sequence length for each byte-level document is 4k and the model processes two documents in parallel each time.

**Architecture.**   On all tasks, the models have 2 layers, with embedding dimension $d = 64$, head number $h = 2$, FFN hidden dimension 128, smaller than those from [20]. Same as [20], we add a CLS token as a global token and use its embedding in the last layer for classification. We re-implement the methods evaluated by Xiong et al. [18], and report the best results of our re-implementation and those reported by Xiong et al. [18]. For our method, the results we run a grid search on the window size $w$ and the projected dimension $r$, and keep $2w + r \leq 256$ to make the complexity similar to the other methods. The maximum sequence length for **ListOps** and **Text** are 2048 and 4096. For **Retrieval**, we set the max sequence for each of the two documents to 4096.

Table 7:  Configurations of our method corresponding to the best results (Transformer-LS (best)) in Table 1.

|  | ListOps (2k) | | Text (4k) | | Retrieval (4k) | |
|---|---|---|---|---|---|---|
|  | $w$ | $r$ | $w$ | $r$ | $w$ | $r$ |
| Dynamic Projection | 0 | 4 | 0 | 128 | 0 | 256 |
| Transformer-LS | 16 | 2 | 1 | 1 | 1 | 254 |

**Hyperparameters for Training.**   Our hyperparameters are the same as Nyströmformer [18] unless otherwise specified. Specifically, we follow [18] and use Adam with a fixed learning rate of $10^{-4}$ without weight decay, batch size 32 for all tasks. The number of warmup training steps $T_w$ and total training steps $T$ are different due to the difference in numbers of training samples. For **Retrieval**, we accidentally found using $T_w = 8000$ rather than the default $T_w = 800$ of [18] improves the results for all models we have evaluated. See Table 8 for the configurations of each task.

**Error bars.**   We have already provided the average of 4 runs with different random seeds in Table 1. Here we also provide the standard deviations for these experiments in Table 9.

Table 8: Training Hyperparameters for LRA tasks.

|  | lr | batch size | $T_w$ | $T$ |
|---|---|---|---|---|
| ListOps | $10^{-4}$ | 32 | 1000 | 5000 |
| Text | $10^{-4}$ | 32 | 8000 | 20000 |
| Retrieval | $10^{-4}$ | 32 | 8000 | 30000 |

Table 9: Accuracy (%) and its standard deviation on Long Range Arena (LRA), with the model configurations and sequence length stats (under the dataset names) annotated. All results are averages of 4 runs with different random seeds. Note that, text has the largest variance of length (i.e., 893).

|  | ListOps (888 ± 339) | | Text (1296 ± 893) | | Retrieval (3987 ± 560) | | Average |
|---|---|---|---|---|---|---|---|
| Model | Acc. | FLOPs | Acc. | FLOPs | Acc. | FLOPs | Acc. |
| Full Attention [1] | 37.1 ± 0.4 | 1.21 | 65.4 ± 0.3 | 4.57 | **82.3** ± 0.4 | 9.14 | 61.59 |
| Reformer [31] (2) | 36.4 ± 0.4 | 0.27 | 64.9 ± 0.4 | 0.58 | 78.6 ± 0.7 | 1.15 | 59.99 |
| Linformer [17] ($k$=256) | 37.4 ± 0.4 | 0.41 | 56.1 ± 1.5 | 0.81 | 79.4 ± 0.9 | 1.62 | 57.62 |
| Performer [28] ($r = 256$) | 32.8 ± 9.4 | 0.41 | 65.2 ± 0.2 | 0.82 | 81.7 ± 0.2 | 1.63 | 59.90 |
| Nyströmformer [18] ($l = 128$) | 37.3 ± 0.2 | 0.61 | 65.8 ± 0.2 | 1.02 | 81.3 ± 0.3 | 2.03 | 61.46 |
| Transformer-LS ($w, r = 8, 32$) | **37.5** ± 0.3 | 0.20 | **66.0** ± 0.2 | 0.40 | 81.8 ± 0.3 | 0.80 | **61.77** |
| Dynamic Projection (best) | 37.8 ± 0.2 | 0.15 | 66.3 ± 0.7 | 0.69 | 81.9 ± 0.5 | 2.17 | 61.98 |
| Transformer-LS (best) | **38.4** ± 0.4 | 0.16 | **68.4** ± 0.8 | 0.29 | **82.0** ± 0.5 | 2.17 | **62.90** |

# C   Additional Results on LRA

## C.1   Results on the image-based tasks of LRA

We give the results of our model on the image-based tasks, implemented in PyTorch, in Table 10.

Table 10: Comparing our model (Transformer-LS) with other methods on the image-based tasks of LRA. For the results of other models, we take their highest scores from [18] and [20].

| Model | Transformer-LS | Linformer | Reformer | Performer | Sparse. Trans. | Nystromformer | Full Att. |
|---|---|---|---|---|---|---|---|
| Image | 45.05 | 38.56 | 43.29 | 42.77 | 44.24 | 41.58 | 42.44 |
| Pathfinder | 76.48 | 76.34 | 69.36 | 77.05 | 71.71 | 70.94 | 74.16 |

## C.2   Compare models implemented in JAX

To compare the results with the implementations from the original LRA paper [20], we re-implement our method in JAX and give the comparisons with other methods in Table 11. The accuracies of other methods come from the LRA paper. We evaluate the per-batch latency of all models on A100 GPUs using their official JAX implementation from the LRA paper. Our method still achieves improvements while being efficient enough. We were unable to run Reformer with the latest JAX since JAX has deleted `jax.custom_transforms`, which is required by the Reformer implementation, from its API.[3] Note the relative speedups from the LRA paper are evaluated on TPUs.

# D   Details for Autoregressive Language Modeling

**An example of long-short term attention for autoregressive models.**   We give an illustration for the segment-wise dynamic projection for autoregressive models as discussed in Section 3.3. With the segment-wise formulation, we can first compute the low-rank projection for each segment in parallel, and each query will only attend to the tokens from segments that do not contain the future token or the query token itself. The whole process is efficient and maintain the $O(n)$ complexity, unlike RFA [30] which causes a slow-down in training due to the requirement for cumulative sum. However,

---

[3] https://github.com/google/jax/pull/2026

Table 11: Comparing the test scores and latency of models on LRA, implemented in JAX.

| Model | ListOps | | Text | | Retrieval | |
|---|---|---|---|---|---|---|
| | Acc. | Latency (s) | Acc. | Latency (s) | Acc. | Latency (s) |
| Local Att | 15.82 | 0.151 | 52.98 | 0.037 | 53.39 | 0.142 |
| Linear Trans. | 16.13 | 0.156 | 65.9 | 0.037 | 53.09 | 0.142 |
| Reformer | 37.27 | - | 56.10 | - | 53.40 | - |
| Sparse Trans. | 17.07 | 0.447 | 63.58 | 0.069 | 59.59 | 0.273 |
| Sinkhorn Trans. | 33.67 | 0.618 | 61.20 | 0.048 | 53.83 | 0.241 |
| Linformer | 35.70 | 0.135 | 53.94 | 0.031 | 52.27 | 0.117 |
| Performer | 18.01 | 0.138 | 65.40 | 0.031 | 53.82 | 0.120 |
| Synthesizer | 36.99 | 0.251 | 61.68 | 0.077 | 54.67 | 0.306 |
| Longformer | 35.63 | 0.380 | 62.85 | 0.112 | 56.89 | 0.486 |
| Transformer | 36.37 | 0.444 | 64.27 | 0.071 | 57.46 | 0.273 |
| BigBird | 36.05 | 0.269 | 64.02 | 0.067 | 59.29 | 0.351 |
| Transformer-LS | 37.65 | 0.187 | 76.64 | 0.037 | 66.67 | 0.201 |

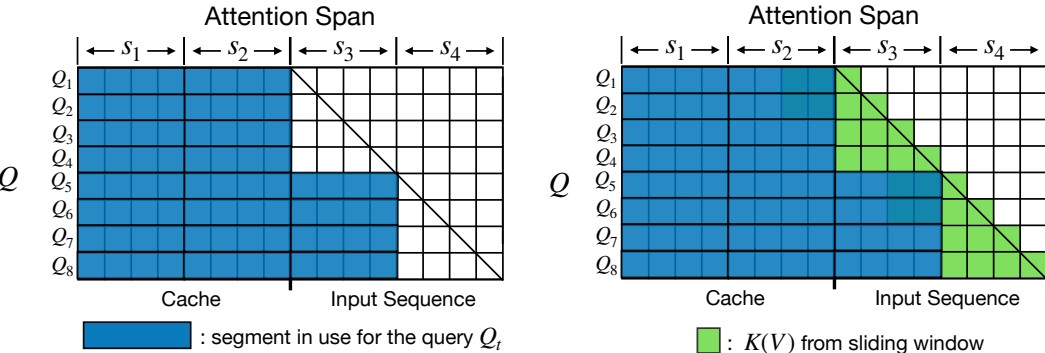

Figure 4: An illustration of effective attention span (colored regions) in Transformer-LS when the segment size for the low-rank attention is $\ell = 4$, and the segment size for the sliding window attention is $w = 2$. Left: the attention span of only the low-rank attention (segment-wise dynamic projection). Right: the attention span of the aggregated attention.

in this way, some of the most recent tokens are ignored, as shown in Figure 4 (left). The window attention (with segment size $w \geq l/2$) becomes an indispensable component in this way, since it fills the gap for the missing recent tokens, as shown in Figure 4.

**Experimental Setup.** Throughout training, we set the window size $w = 512$, the segment length $l = 16$, and the dimension of the dynamic low-rank projection $r = 1$, which in our initial experiments achieved better efficiency-BPC trade-off than using $l = 32, r = 1$ or $l = 64, r = 4$. Our small and large models have the same architecture as Longformer [14], except for the attention mechanisms. We use similar training schedules as Longformer [14]. Specifically, for all models and both datasets, we train the models for 430k/50k/50k steps with 10k/5k/5k linear learning rate warmup steps, and use input sequence lengths 2048/4096/8192 for the 3 phases. We use constant learning rate after warmup. We compared learning rates from {1.25e-4, 2.5e-4,5e-4,1e-3} for 100k iterations and found 2.5e-4 to work the best for both models on enwik8, and 5e-4 to work the best on text8. The batch sizes for the 3 phases are 32, 32, 16 respectively. Unlike Longformer and Transformer-XL, we remove gradient clipping and found the model to have slightly faster convergence in the beginning while converging reliably. For smaller models, we use dropout rate 0.2 and weight decay 0.01. For the larger model, we use dropout 0.4 and weight decay 0.1.

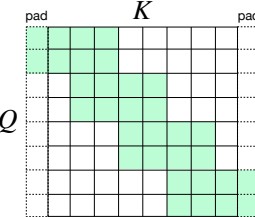 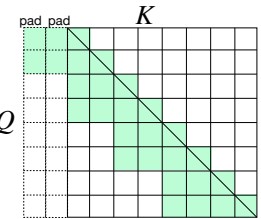

For Bidirectional Models    For Autoregressive Models

Figure 5: An illustration of our sliding window attention in 1D autoregressive and bidirectional models. Here, we use a group size $w = 2$. Each token inside each group are restricted to attend to at most $2w$ tokens. In the bidirectional model, they attend to $w$ tokens from the home segment, and $w/2$ tokens to the left and right of the home segment respectively. In the autoregressive model, they attend to $w$ tokens to the left of the home segment, as well as all tokens within the home segment that is not a future token.

# E   Details for ImageNet Classification

**The CvT Architecture.**    We implement the CvT model based on a public repository, [4] because this is a concurrent work with no official implementation when we conduct this work. In Table 5, since our CvT re-implementation gets worse test results than reported ones in their arxiv paper, we still list the best test accuracy from Wu et al. [6] for fair comparisons. We report the FLOPs of CvT with our implementation for reasonable comparisons, because our CvT*-LS implementation is based on that. Same as CvT, all the models have three stages where the first stage downsamples the image by a factor of 4 and each of the following stages downsamples the feature map by a factor of 2. CvT*-LS-13 and CvT*-LS-21 have the same configuration as CvT-13 and CvT-21. CvT*-LS-17 and CvT*-LS-21 are our customized models with more layers and higher embedding dimensions in the first two stages ($[3, 4, 10], [3, 4, 14]$ layers respectively and $[128, 256, 768]$ dimensions). We train the model for 300 epochs using a peak learning rate of $5e - 4$ with the cosine schedule [65] with 5 epochs of warmup. We use the same set of data augmentations and regularizations as other works including PVT [5] and ViL [11]. In general, CvT*-LS-13 and CvT*-LS-21 closely follow the architectural designs of CvT for fair comparisons. Specifically, in CvT*-LS, we feed the token embeddings extracted by the depth-wise separable convolution [66] of CvT to our long-short term attention. For dynamic projection, we replace $W_i^P$ in Eq. (3) with a depth-wise separable convolution to maintain consistency with the patch embeddings, but we change its BN layer into a weight standardization [67, 68] on the spatial convolution's weights for simplicity. We do not use position encoding. All of our models have 3 stages, and the feature map size is the same as CvT in each stage when the image resolutions are the same. CvT*-LS-13 and CvT*-LS-21 follow the same layer configurations as CvT-13 and CvT-21, i.e., the number of heads, the dimension of each head and the number of Transformer blocks are the same as CvT in each stage. For all models on resolution $224 \times 224$, we set $r = [64, 16, 4]$ and $w = [8, 4, 2]$. For higher resolutions, we scale up $r$ and/or $w$ to maintain similar effective receptive fields for the attentions. At resolution $384 \times 384$, we use $r = [64, 16, 4]$ and $w = [12, 6, 3]$ for the 3 stages. At resolution $448 \times 448$, we use $r = [128, 32, 8]$ and $w = [16, 8, 4]$.

Besides maintaining the CvT architectures, we also try other architectures to further explore the advantage of our method. With the efficient long-short term attention, it becomes affordable to stack more layers on higher-resolution feature maps to fully utilize the expressive power of attention mechanisms. Therefore, we have created two new architectures, CvT*-LS-17 and CvT*-LS-21S, that have more and wider layers in the first two stages, as shown in Table 12. Compared with CvT-21, CvT*-LS-17 has 25% fewer parameters, less FLOPs, but obtained the same level of accuracy. CvT*-LS-21S has fewer parameters than CvT*-LS-21, more FLOPs, and 0.4% higher accuracy, demonstrating the advantage of focusing the computation on higher-resolution feature maps.

**The effect of DualLN.**    We trained the CvT*-LS-13 model without DualLN, which has a test accuracy of 81.3, lower than the 81.9 with DualLN.

---

[4]https://github.com/rishikksh20/convolution-vision-transformers

Table 12: Architectures of our CvT*-LS-17 and CvT*-LS-21S models. LSTA stands for our Long-Short Term Attention.

|  | Output Size | Layer Name | CvT*-LS-17 | CvT*-LS-21S |
|---|---|---|---|---|
| Stage 1 | $56 \times 56$ | Conv. Embed. | $7 \times 7, 128$, stride 4 | |
|  | $56 \times 56$ | Conv. Proj. LSTA MLP | $\begin{bmatrix} 3 \times 3, 128 \\ H = 2, D = 128 \\ r = 64, w = 8 \\ R = 4 \end{bmatrix} \times 3$ | |
| Stage 2 | $28 \times 28$ | Conv. Embed. | $3 \times 3, 256$, stride 2 | |
|  | $28 \times 28$ | Conv. Proj. LSTA MLP | $\begin{bmatrix} 3 \times 3, 256 \\ H = 4, D = 256 \\ r = 16, w = 4 \\ R = 4 \end{bmatrix} \times 4$ | |
| Stage 3 | $14 \times 14$ | Conv. Embed. | $3 \times 3, 384$, stride 2 | |
|  | $14 \times 14$ | Conv. Proj. LSTA MLP | $\begin{bmatrix} 3 \times 3, 384 \\ H = 6, D = 384 \\ r = 4, w = 2 \\ R = 4 \end{bmatrix} \times 10$ | $\begin{bmatrix} 3 \times 3, 384 \\ H = 6, D = 384 \\ r = 4, w = 2 \\ R = 4 \end{bmatrix} \times 14$ |

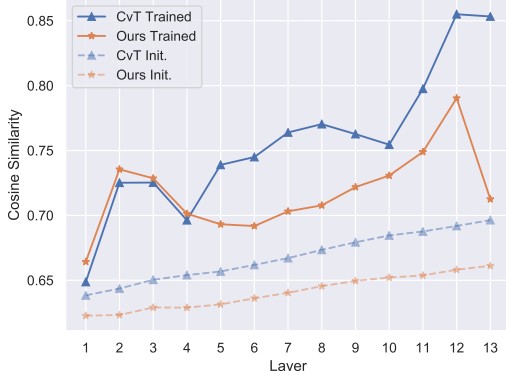

Figure 6: Pairwise cosine similarity between patch embeddings at different layers of CvT-13 and CvT*-LS-13, averaged on 50k images of ImageNet validation set. The larger cosine similarities at deeper layer suggest that the feature representation is less diverse.

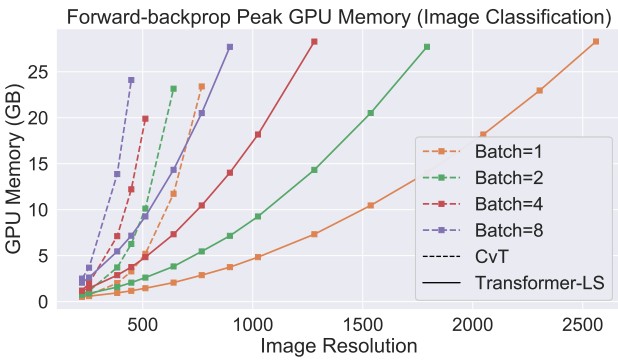

Figure 7: Running memory consumption of full self-attention (CvT-13) and Long-Short Transformer on different tasks. We increase the sequence length resolution until the model is out of memory on a V100 GPU with 32GB memory.

# F   Evaluate the robustness of models trained on ImageNet-1k.

Table 13: Corruption Error (CE) on ImageNet-C

| Arch. | Noise | | | Blur | | | | Weather | | | | Digital | | | |
| | Gauss. | Shot | Impulse | Defocus | Glass | Motion | Zoom | Snow | Frost | Fog | Bright | Contrast | Elastic | Pixel | JPEG |
|---|---|---|---|---|---|---|---|---|---|---|---|---|---|---|---|
| ResNet-50 | 34.24 | 49.25 | 55.84 | 56.24 | 57.04 | 63.53 | 63.68 | 64.02 | 64.04 | 64.89 | 69.25 | 70.72 | 73.14 | 75.29 | 75.76 |
| DeiT-S | 26.93 | 36.81 | 36.89 | 39.38 | 40.14 | 43.32 | 43.80 | 44.36 | 45.71 | 46.90 | 47.27 | 48.57 | 52.15 | 57.53 | 62.91 |
| CvT*-LS-13 | 25.64 | 36.89 | 37.06 | 38.06 | 43.78 | 43.78 | 44.62 | 45.92 | 47.77 | 47.91 | 49.60 | 49.66 | 54.92 | 57.24 | 68.72 |
| CvT*-LS-17 | 25.26 | 35.06 | 35.48 | 37.38 | 41.37 | 43.95 | 44.47 | 46.05 | 46.17 | 46.38 | 49.08 | 49.37 | 54.29 | 54.54 | 69.54 |
| CvT*-LS-21 | 24.28 | 34.95 | 35.03 | 35.93 | 39.86 | 40.71 | 41.27 | 41.78 | 44.72 | 45.24 | 45.50 | 47.19 | 51.84 | 53.78 | 67.05 |

Table 14: Robustness evaluation on ImageNet-9. We report Top-1 Accuracy.

| Model | Params (M) | ImageNet (%) | ImageNet-9 [59](%) | | |
| | | | Original | Mixed-same | Mixed-rand |
|---|---|---|---|---|---|
| ResNet-50 [35] | 25.6 | 76.2 | 94.9 | 87.1 | 81.6 |
| DeiT-S [36] | 22.1 | 79.8 | 97.1 | 89.1 | 84.2 |
| CvT*-LS-13 | 20.3 | 81.9 | 97.0 | 90.7 | 85.6 |
| CvT*-LS-21 | 32.1 | **82.7** | **97.2** | **91.5** | **85.8** |

For a fair comparison, we choose models with similar number of parameters. We select two representative models, including the CNN-based model (ResNet) and the transformer-based model (DeiT). We give detailed results on all types of image transforms on ImageNet-C in Table 13. We evaluate our method on various ImageNet robustness benchmarks as follows:

- **ImageNet-C**. ImageNet-C refers to the common corruption dataset. It consists of 15 types of algorithmically common corruptions from noise, blur, weather, and digital categories. Each type contains five levels of severity. In Table 4, we report the normalized mean corruption error (mCE) defined in Hendrycks and Dietterich [56]. In Table 13, we report the corruption error among different types. In both tables, the lower value means higher robustness.

- **ImageNet-A**. ImageNet-A is the natural adversarial example dataset. It contains naturally collected images from online that mislead the ImageNet classifiers. It contains 7,500 adversarially filtered images. We use accuracy as our evaluation metric. The higher accuracy refers to better robustness.

- **ImageNet-R**. ImageNet-R (**R**endition) aims to evaluate the model generalization performance on out-of-distribution data. It contains renditions of 200 ImageNet classes (e.g. cartoons, graffiti, embroidery). We use accuracy as the evaluation metric.

- **ImageNet-9**. ImageNet-9 aims to evaluate the model background robustness. It designs to measure the extent of the model relying on the image background. Following the standard setting [59], we evaluate the two categories, including MIXED-SAME and MIXED-RAND. MIXED-SAME refers to replace the background of the selected image with a random background of the same class by GrabCut [59]; MIXED-RAND refers to replace the image background with a random background of the random class.

From table 6, we find that our method achieves significant improvement compared to CNN-based network (ResNet). For instance, our method improves the accuracy by 23.6%, 22.1%, 9.7% compared to ResNet on ImageNet-C, ImageNet-A, and ImageNet-R, respectively. For ImageNet-9, our method also achieves favorable improvement by 4.3% on average (Mixed-same and Mixed-rand). It indicates that our method is insensitive to background changes. We guess the potential reasons for these improvements are (1) the attention mechanism and (2) the strong data augmentation strategies during the training for vision transformer [4, 36]. The first design helps the model focus more on the global context of the image as each patch could attend to the whole image areas. It reduces the local texture bias of CNN. The latter design increases the diversity of the training data to improve model's generalization ability. Compared to DeiT, we also surprisingly find that our method achieves slightly better performance. One plausible explanation is that our long-term attention has a favorable smoothing effect on the noisy representations. Such improvements also indicate that different designs of attention and network architecture can be essential to improve the robustness. As the goal of this paper is not to design a robust vision transformer, the robustness is an additional bonus of our method.

We believe that our observation opens new directions for designing robust vision Transformers. We leave the in-depth study as an important future work.

The detailed results of ImageNet-C and ImageNet-9 are shown in Table 13 and Table 14 respectively.