# OpenReview forum: "Long-Short Transformer: Efficient Transformers for Language and Vision"
_NeurIPS.cc/2021/Conference — NeurIPS 2021 Poster_

### Official Review · Reviewer_K8ya · 2021-07-12

**Rating:** 6
**Confidence:** 4

**Summary:**

This paper proposed Long-Short Transformer (Transformer-LS), which aggregates sliding window attention (i.e. short-term attention) with linear attention via low-rank projection (i.e. long-term attention), by simply concatenating the two sets of keys.
Due to the scale mis-match of the keys from the long and short term attentions, the authors further proposed the dual layernorm (DualLN) method.

Experiments were conducted on Long-range Arena (LRA), auto-regressive language modeling and ImageNet classification. Transformer-LS achieves improvements over previous efficient transformer models on both accuracy and efficiency.

**Main Review:**

The paper is clear and generally well written.

However, there are some weaknesses:

1. The proposed architecture of Transformer-LS is not novel. The two key components of Transformer-LS, i.e. the sliding window attention and low-rank projection (or similar architectures), have been introduced in existing works. For example, the sliding window attention has been extensive explored in LongFormer and Big-Bird models (with some slight differences), and the low-rank projection approach is a straight-forward extension of LinFormer. Actually, a recent work (Goyal et al., 2021) proposes a more general low-rank projection approach which captures contextual information in the projection, instead of just using a learnable parameter (like W_i^{P} in E.q 3). Thus, the contribution of this work is to combine the two types efficient attention mechanisms and empirically evaluate it on language and vision tasks.


2. The experimental results, especially  those on language tasks, are not entirely convincing.
For the experiments on LRA, they did not follow the default setting in the LRA paper, and the efficiency comparison is indirect. It is highly recommended to follow the default setting in the LRA paper to fairly compare the accuracy with previous efficient transformer models, and directly compare the speed and memory efficiency by steps per second and memory usage.
Moreover, it is highly recommended to conduct more challenging language tasks, including machine translation and pre-trained language models (such as BERT), to demonstrate the effectiveness of Transformer-LS.

References:

Coordination Among Neural Modules Through a Shared Global Workspace. Goyal et al., 2021.

================Updates after reading authors' response=============================

Thanks for the authors' great effort on the response, which mostly addressed my concerns. Thus, I upgraded my score to 6.

**Time Spent Reviewing:**

3

---

> ### Author Response · Authors · 2021-08-10
> **Response to Reviewer K8ya**
>
> Many thanks for your review. We will address your comments in the following.
>
> 1. “The proposed architecture of Transformer-LS is not novel. The two key components of Transformer-LS, i.e. the sliding window attention and low-rank projection (or similar architectures), have been introduced in existing works.  For example, the sliding window attention has been extensive explored in LongFormer and Big-Bird models (with some slight differences), and the low-rank projection approach is a straight-forward extension of LinFormer … Thus, the contribution of this work is to combine the two types efficient attention mechanisms and empirically evaluate it on language and vision tasks.”
> - We disagree that combining existing attention mechanisms simply makes this work not novel. Both the sliding window attention and  (some types of) low-rank projection have been studied in previous work respectively, but each component alone will not provide the state-of-the-art results. In fact, the combination of different types of attention mechanisms is the key for empirical success in the domain. For example, Longformer combines sliding window attention and task-motivated global token attention. Big-Bird further combines them and random attention. Indeed, from the NeurIPS review guideline of Originality, it states that the work with a novel combination of well-known techniques  can be valuable. Besides, our dynamic low-rank projection is novel and has not been studied for long sequence modeling before.
> - In this work, we propose a novel combination of sliding window and low-rank attentions, which obtains the state-of-the-art results for various NLP and vision tasks with linear complexity w.r.t input length. Indeed, simply combining these two attentions together will not provide satisfactory results. Specifically, we have proposed two non-trivial technical contributions to achieve the good results, including 1) input-dependent dynamic low-rank projection as the long-range attention; 2) DualLN to handle the scale mismatch problem when integrating long-range and short-term attentions. We provide ablation studies to show that all of the above three contributions are essential for the state-of-the-art results (see the details in the Ablation Studies at the end). We believe such empirical success and the insights for improving the results are as valuable as proposing a brand new attention mechanism.
>
> 2. "a recent work (Goyal et al., 2021) proposes a more general low-rank projection”
> - Thank you for pointing it out. We will cite Goyal et al., 2021 and discuss it accordingly in the final version of our paper. To our understanding, although this paper also focuses on attention mechanisms, the methods are not the same, and it is evaluated on a completely different set of tasks.
>
> 3. “For the experiments on LRA, they did not follow the default setting in the LRA paper, and the efficiency comparison is indirect. It is highly recommended to follow the default setting in the LRA paper to fairly compare the accuracy with previous efficient transformer models, and directly compare the speed and memory efficiency by steps per second and memory usage.”
> - We have provided the code in the supplementary for sanity check.
> - We believe we are providing fair comparisons with other models. We use the same code to run these models and report the best results among our reimplementation and the results from the LRA and Nystromformer paper. We achieve these results at no additional FLOPs.
> - We use the same datasets from the official LRA repository. We use PyTorch for the implementation of our method and all previous efficient transformer models, which follows the same setting as previous work (e.g., Xiong et al.). Note that we do find the PyTorch implementations provide slightly higher accuracies than results reported in the LRA paper across baseline models on ListOps, Text and Retrieval, but all the reported LRA results from Xiong et al.  are using the same PyTorch implementations (https://github.com/mlpen/Nystromformer). As a result, our comparisons are fairly conducted.
>
> Xiong et al. Nystromformer: A Nystrom-based Algorithm for Approximating Self-Attention.
>
> 4. “conduct more challenging language tasks, including machine translation and pre-trained language models (such as BERT)”
> - Thank you for your suggestion. During this short period of time for rebuttal, we apply our method to pretrained language models, based on RoBERTa-base and RoBERTa-large using the same training schedule as Longformer. We finetune the pretrained models on IMDb and obtained better results than RoBERTa and Longformer. We plan to further verify its efficacy on QA tasks in the future. We will add the results in the final version of our paper.
>
> | Model               |  RoBERTa-base | RoBERTa-large | Longformer-base | LS-base     | LS-large |
> | ----------------      |  ------------           |  ---------------       |   -----------------   |  ---------------  |  ------------ |
> | IMDb Accuracy |     95.3               |     96.5               |        95.7           |   96.0             |   96.8       |
>
>
> &nbsp;
>
> ## Ablation Studies
>
> Specifically, we conduct ablation studies as follows to show that all of the above two contributions are the key for empirical success.
>
> - Contribution 1:  dynamic low-rank projection in long-range attention
>
>   In Table 1 of our submission for LRA, when we compare the results comparing Linfromer and Dynamic Projection. We could find that Dynamic Projection could help to achieve non-trivial improvement (e.g. Dynamic Projection achieves 66.28 Acc compared to 56.12 of Linformer in Text). The Text dataset has the highest variance in sequence length among the 3 tasks (see Table 7 in Appendix for a comparison), so the advantage of the flexibility of Dynamic Projection is the most significant.
>
>   Additionally, we also add two more experiments including (1) Linformer + Window attention and (2) Dynamic Projection + Window attention, to show the effectiveness of Dynamic projection in the local + global window attention mechanism. The settings are the same as in Section 4.1.
>
> |  Task                    |  ListOps  |   Text |  Retrieval  |
> | -------                    |  ------       |    ----  |  ---------     |
> | Linformer + Win.  | 38.03      | 59.63 | 79.68        |
> | Dynamic Projection + Win. |  38.36     | 68.40 | 81.95        |
>
>   We could find that dynamic projection consistently improves the performance on these three LRA tasks.  It further verifies the importance of dynamic projection.
>
>   We  also conducted the experiment based on the ViL-Medium model on ImageNet. The results are as follows.
>
>     Linformer  + Win: 83.0 %
>     Dynamic Projection + Win: 83.2%
>   From this result, we also observe that the results with dynamic projection are higher than the results with Linformer. It further verifies that dynamic projection is indeed an important component to improve the performance on ImageNet.
>
>   Moreover, as we mentioned in the paper,  our Dynamic Projection is more robust to positional changes like insertion and perturbation. In the table below, we also show the results of the robustness of  Dynamic Projection against the positional changes like insertion and Deletion.
>
> |  Task      |    Text |  Text (Insertion) | Text (Deletion) | Retrieval | Retrieval (Insertion) | Retrieval (Deletion) |
> | ----------- |    -----  | ----------------    | ----------------        |   ----------  | ----------------            | ---------------- |
> | Linformer     | 56.12 | 55.94 |   54.91            | 79.37       | 53.66          | 51.75 |
> | DP               | 66.28 | 63.16 |   58.95            | 81.86       | 70.01          | 64.98  |
> | Linformer + Win.   | 59.63 | 56.69  | 56.29     | 80.01       | 52.83          | 52.13  |
> | DP + Win. (ours) | 68.40 | 66.34  | 62.62     | 81.95       |  69.93         | 64.19  |
>
>   These results show the robustness of the Dynamic Projection.
>
>
> - Contribution 2:  DualLN to handle the norm mismatch problem when integrating long-range and short-term attention.
> To show the effectiveness of DualLN, we also gave the comparisons in Figure 1 and Line 622 (appendix). In Figure 1, we have shown that without DualLN, the sparse and low-rank embeddings have a magnitude mismatch. With DualLN, the ratios will be 1.0 at every layer, which will facilitate optimization. Figure 1 right, we also show that the Transformer-LS models trained with DualLN have consistently lower validation loss than the models without DualLN.
>
>   We have also added experiments based on the ViL-Medium model on ImageNet, we show DualLN improves ViL-LS-Medium’s accuracy from 83.0 to 83.2.

---

> > ### Comment · Reviewer_K8ya · 2021-08-11
> > **Questions about experimental setting on LRA and autoregressive attention**
> >
> > Thanks for the authors' effort on the response, which mostly addressed my concerns. But I am still concerned about the setting of the experiments on LRA. Why not following the default setting in the official LRA paper? LRA paper provided results of many efficient Transformer models as baselines for comprehensive comparison. Moreover, why not reporting the relative speed up and peak memory cost in the same way as the LRA paper, but only reporting FLOPs? Since each Transformer-LS layer contains two attention operations, it is highly recommended to compare speed and memory with other efficient Transformer models.
> >
> > My second question is about the implementation of the causal attention. The authors, judging from their submitted code, do not seem to implement the auto-regressive attention using CUDA kernels. Due to the sequential computation, what is the training speed comparison for autoregressive tasks such as language modeling?

---

> > > ### Author Response · Authors · 2021-08-17
> > > **LRA results in JAX, and clarifications about autoregressive attention**
> > >
> > > Many thanks for your reply. We are glad to see our response mostly addressed your concerns. We will answer your remaining questions in the following.
> > >
> > > 1, “Why not following the default setting in the official LRA paper? ”
> > > - Our implementation on LRA is based on a well-accepted PyTorch implementation from the Nystromformer paper (https://github.com/mlpen/Nystromformer), which is accepted by AAAI and has achieved state-of-the-art results before our paper. A minor reason for us not to use the implementation of the LRA paper is that it is written in JAX, but we do not have expertise in JAX and all our implementations were based on PyTorch.
> > > - We emphasize that our comparisons with other Transformer models in Table 1 are fair. All the other Transformers we compared in the paper use the same implementation and same hyperparameters (lr, iterations, schedule, number of attention heads, hidden dimensions, number of layers, etc.) except for their attention-specific parameters (e.g., our $w,r$, the $l$ of Nystromformer, see Table 1). Under these constraints, we obtained better accuracy with fewer FLOPs. We believe this demonstrates the efficacy and efficiency of our method.
> > > - To show that our method maintains its advantage in a different implementation, we have re-implemented it in JAX based on https://github.com/google-research/long-range-arena. We keep all the default configurations unchanged from this official implementation and only set the $w,r$ of our method. We also compare the per-batch latency under the default batch sizes for other methods on the same machine.
> > >   - **The accuracy and speed comparisons are given below. The accuracies of other methods come from the LRA paper, but we test the per-batch latency of all models on A100 GPUs using the official JAX implementation. Our method still achieves improvements while being efficient enough, despite all the difficulties with the JAX implementations.** We were unable to run Reformer with the latest JAX since JAX has deleted `jax.custom_transforms` (required by the Reformer implementation) from its API (https://github.com/google/jax/pull/2026) and we were unable to configure its substitute (custom_jvp/vjp) correctly. Note the relative speedups from the LRA paper are evaluated on TPUs which we do not have.
> > > | Model           | ListOps |   ListOps   |  Text |     Text    | Retrieval |  Retrieval  |
> > > |-----------------|:-------:|:-----------:|:-----:|:-----------:|:---------:|:-----------:|
> > > |                 |   Acc.  | latency (s) |  Acc. | latency (s) |    Acc.   | latency (s) |
> > > | Local Att       |  15.82  |    0.151    | 52.98 |    0.037    |   53.39   |    0.142    |
> > > | Linear Trans.   |  16.13  |    0.156    | 65.90 |    0.037    |   53.09   |    0.142    |
> > > | Reformer        |  37.27  |      -      | 56.10 |      -      |   53.40   |      -      |
> > > | Sparse Trans.   |  17.07  |    0.447    | 63.58 |    0.069    |   59.59   |    0.273    |
> > > | Sinkhorn Trans. |  33.67  |    0.618    | 61.20 |    0.048    |   53.83   |    0.241    |
> > > | Linformer       |  35.70  |    **0.135**    | 53.94 |    **0.031**    |   52.27   |    **0.117**    |
> > > | Performer       |  18.01  |    0.138    | 65.40 |    **0.031**    |   53.82   |    0.120    |
> > > | Synthesizer     |  36.99  |    0.251    | 61.68 |    0.077    |   54.67   |    0.306    |
> > > | Longformer      |  35.63  |    0.380    | 62.85 |    0.112    |   56.89   |    0.486    |
> > > | Transformer     |  36.37  |    0.444    | 64.27 |    0.071    |   57.46   |    0.273    |
> > > | BigBird         |  36.05  |    0.269    | 64.02 |    0.067    |   59.29   |    0.351    |
> > > | Ours            |  **37.65**  |    0.187    | **76.64** |    0.037    |   **66.67**   |    0.201    |
> > >
> > >  - Note that JAX does not support `as_strided` as in NumPy and PyTorch (https://github.com/google/jax/issues/3171), which we used to improve the efficiency our window attention, just like the `Longformer-chunk` implementation in the Longformer paper. **Therefore, the speed of our window attention is probably compromised in JAX.** The speed comparison of Longformer was excluded from the LRA paper, probably due to their less-efficient implementation of the window attention in JAX.
> > >   - We cannot provide an accurate characterization of memory usage in JAX since it relies on an internal tool that has not been open-sourced yet (https://github.com/google-research/long-range-arena/issues/32). JAX takes up 90% of the memory by default or it may cause additional memory usage in other modes (https://jax.readthedocs.io/en/latest/gpu_memory_allocation.html). **In general, our method fits into one single A100 GPU on all the 3 tasks. However, Sparse Transformer, Synthsizer, Longformer and Transformer (full attention) need 2 and 4 A100 GPUs respectively on ListOps and Retrieval.**
> > >
> > >
> > >
> > > 2, “My second question is about the implementation of the causal attention. The authors, judging from their submitted code, do not seem to implement the auto-regressive attention using CUDA kernels. Due to the sequential computation, what is the training speed comparison for autoregressive tasks such as language modeling?”
> > > - We did not upload the code for causal attention since we did not manage to clean the code before the supplementary deadline. We will release our code and models for auto-regressive attention. Note that, one does not have to write CUDA kernel for efficient implementation with linear complexity. Same as the implementation of `Longformer-chunk` from the Longformer paper, we use `torch.as_strided` to construct a view of the key and query tensors for the window segments without copying. We mask out the future tokens in each segment as shown in Figure 3 in Appendix. This implementation is faster than the CUDA kernel while maintaining a linear memory complexity (See the comparison between `Longformer-chunk` and `Longformer-cuda` in Figure 1 of Longformer paper).
> > > - **In Figure 2 of our submission, we have provided the comparison of the training speed and memory consumption with Transformer-XL (based on Huggingface's implementation) on autoregressive language modeling.** The official implementation of autoregressive Longformer on this task is not available so we did not compare with it.

---

### Official Review · Reviewer_8e8m · 2021-07-15

**Rating:** 7
**Confidence:** 3

**Summary:**

In this paper, the authors focus on reducing the quadratic computation complexity of the self-attention mechanism in the standard Transformer. They combine local and global context through sliding window attention and matrix projections. During aggregating long-range and short-term attention, they employ two sets of Layer Normalizations to make sure that the model will use both long and short range effectively. Experiments show some promising results on various tasks.

**Limitations And Societal Impact:**

 Although they do not explicitly mention the societal impact, I think their method could help us to build the efficient Transformer. However, I still suggest authors add this section to their draft.

**Main Review:**

Their model tries to model both the long-range and fine-grained local correlations using two different attentions, both of which have linear computation complexity. Their method is easy to understand but I do think it is not rival to implement this model so it would be great if authors could release their code to the public. Their paper is easy to follow and extensive experiments has shown the effectiveness of their method. I just have a few questions:
1. For aggregating the local and short-range attention, you concatenate their attention matrix. Will this introduce extra computation, could you just simply add them together, and will this hurt the performance much?
2. Will you apply your method for the pre-trained language model to further demonstrate the effectiveness and efficiency of your proposed method?
3. For efficiency, FLOPs are used to compare different models, and could you compare their wall time speed?

**Time Spent Reviewing:**

1.5

---

> ### Author Response · Authors · 2021-08-10
> **Response to Reviewer 8e8m**
>
> Thank you so much for your review and suggestions. They are really helpful for improving our paper.
>
> 1. “It would be great if authors could release their code to the public.”
> - Many thanks. We will release our code and models.
>
> 2. Will the concatenation introduce extra computation? Could you just simply add local and global attentions together, and will this hurt the performance much?
> - The concatenation does not cause significant overhead. For implementation, concatenation only happens when computing the softmax scores over the keys. We first compute the inner product between the keys and queries from the window and global attention separately, and then concatenate the inner products, which is just concatenating a $n\times 2w$ and a $n\times r$ matrix. $w$ and $r$ are much smaller than the sequence length $n$ and the feature dimensions $d$, so this operation does not cause significant overhead.
> - We also tried adding the window and global attentions in our preliminary experiments, but the results were not as good. We plan to include more detailed comparisons in our future version.
>
> 3. Will you apply your method for the pre-trained language model to further demonstrate your proposed method?”
> - Yes, we also apply our method to pretrained language models, based on RoBERTa-base and RoBERTa-large using the same training schedule as Longformer. We finetune the pretrained models on IMDb and obtained better results than RoBERTa and Longformer. We plan to further verify its efficacy on QA tasks in the future. We will add the results in the final version of our paper.
>
> | Model               |  RoBERTa-base | RoBERTa-large | Longformer-base | Transformer-LS-base | Transformer-LS-large |
> | ----------------      |  ------------           |  ---------------       |   -----------------   |  ---------------  |  ------------ |
> | IMDb Accuracy |     95.3               |     96.5               |        95.7           |   96.0             |   96.8       |
>
> 4. “For efficiency, FLOPs are used to compare different models, and could you compare their wall time speed?”
> - Yes. We will add latency results as follows in the final version of this paper.
>
> Latency is evaluated with batch size 32.
>
>  |  Model           | Resolution |  Top-1 Acc.  |    Params (M)  |  FLOPs (G)  |   latency (s)  |
>  | ----------------   |  :------------:  |  :---------------: |   :--------------:  |  :---------------:  |  :---------------: |
>  | ViL-Medium  |    $224\^2$  |  83.5            |   39.7                 |   8.7             |     0.106      |
>  | ViL-Base       |  $224\^2$   |   83.7            |   55.7                 | 13.4             |     0.164      |
>  | ViL-LS-Medium  |    $224\^2$  |  83.8            |   39.8                  |   8.7             |     0.075      |
>  | ViL-LS-Base        |   $224\^2$   |   84.1           |   55.8                  | 13.4             |     0.113      |
>  | ViL-LS-Medium  |    $384\^2$  |  84.4            |   39.8                  |   28.7            |    0.271     |
>
> “Limitations And Societal Impact”
> - We mentioned the limitations and social impact in the Checklist 1.(c), and included the following discussion in Appendix.
> > Transformer-based methods have achieved great successes in many domains including vision and language. However, the computational and memory complexity of the full attention mechanism scales quadratically with the length of the input. Such inefficiency introduces a growing concern in the ML community about the resource and energy requirement to train large-scale systems. Therefore, in this paper, we build computationally efficient systems that can capture long sequence input in an energy-efficient way without losing expressive power.
> > All datasets used in our paper are publicly available. To the best of our knowledge, the datasets do not contain any personally identifiable information. Like many other methods, our model is data-driven and has the risk of being biased. We encourage careful consideration before deploying Transformer-LS to sensitive applications that have ethical implications.

---

### Official Review · Reviewer_J3tL · 2021-07-15

**Rating:** 6
**Confidence:** 4

**Summary:**

The paper introduces Long-Short Transformer, an efficient Transformer model that combines local sliding window attention with low rank global attention. This is an attempt in combining the two dominant paradigms of making Transformers more efficient: Sparsifying attention and Low-rank methods.

**Limitations And Societal Impact:**

I would have liked to see some discussion on limitations of the work and societal impacts.
Some that come to mind are potential environmental impacts of training and experimenting with such models at scale (e.g., https://arxiv.org/pdf/1907.10597.pdf).

**Main Review:**

There are two main ideas presented in the paper: 1- Combining sparse local attention with low-rank methods. 2- Using dynamic low-rank projection with a function that depends on the Keys instead of a fixed learned projection.
The general ideas are very interesting and results are encouraging. However, there are few key weaknesses with the paper:

First, while I found the first idea to be very interesting, the impact of the second idea is less clear as I couldn't find any ablations comparing the full model with the model that combines the local attention with a fixed low-rank projection methods. I.e., What would happen if we replace the dynamic projection with fixed learned projection (similar to the Linformer paper)?
Similarly, the impact of the local sliding window is also not very clear, what would happen if the queries only attended to global projections? The only place such results exists appears to be on the 3 Long Range Arena tasks. Another useful ablation would have been comparing dynamic projection with fixed projection in the setting without the addition of local sliding window attention.

The second weakness is the general evaluation tasks. I appreciate the experiments on LRA and char language modeling, but there is no evaluation on actual downstream NLP tasks, which makes the effectiveness of the model in real-world applications less clear. Instead of expensive char LM experiments, I would have much preferred experiments on NLP tasks similar to those evaluated by Longformer and Big Bird papers.

I was a bit disappointed to see some of the tasks of LRA excluded from evaluations. The authors mention that this is because the data is small and tasks are synthetic, but in general other LRA tasks are also not realistic and I was a bit confused about this evaluation setup. About the size of data, does this mean that the model will have harder time training in low-resource setting?

On Robustness results (Table 4), how does a CvT model (without the addition of Long Short Transformer) work? This comparison seems to be missing.

The paper mentions that dynamic projections are "more flexible and robust to, e.g., insertion, deletion, paraphrasing, and other operations that change sequence length" (L164-165). However, I didn't find any empirical evidence or analyses about this.

Minor: A figure about the main idea would have been nice to include.

Update: The authors included new experiments addressing most of the above concerns. I'm updating my score.




**Time Spent Reviewing:**

5

---

> ### Author Response · Authors · 2021-08-10
> **Response to Reviewer J3tL**
>
>
> Many thanks for your constructive comments and acknowledging that our idea is interesting and results are encouraging. We address the comments as follows.
>
> 1. “ablations comparing the full model with the model that combines the local attention with a fixed low-rank projection methods.”
> - This is a very good question. Based on your suggestion, we conduct additional experiments by comparing local attention with a fixed low-rank projection (Linformer + Win. ) and local attention with dynamic project ( Dynamic Projection + Win.) on LRA tasks and ImageNet task. The results are shown as follows.
>     -  We compare these two methods on LRA. The settings are the same as in Section 4.1.
> |  Task                    |  ListOps  |   Text |  Retrieval  |
> | -------                    |  :------:     |    :----:  |  :---------:     |
> | Linformer + Win.                 | 38.03      | 59.63 | 79.68        |
> | Dynamic Projection + Win. |  38.36     | 68.40 | 81.95        |
>
>     - ImageNet. We conduct the experiment based on the ViL-Medium model on ImageNet with 150 epochs due to the limited time of rebuttal period. The results are as follows:
>         - Linformer  + Win: 83.0 %
>         - Dynamic Projection + Win: 83.2 %
>
>     We observe that the results with Dynamic Projection + Win are always higher than the results with Linformer + Win. It further verifies that dynamic projection is indeed an important component to improve the performance.
>
>
>  2. “The impact of the local sliding window. what would happen if the queries only attended to global projections? ”
> - Note we have given the comparisons of LRA in Table 1 (Linformer, Dynamic Projection, Transformer-LS). From the results, we observe that Transformer-LS achieves a higher performance than Linformer and Dynamic Projection. It shows the importance of our local + global mechanism. Additionally,  we could observe that Dynamic projection has consistently  achieved higher performance compared to Linformer (sole fixed projection). It also verifies the importance of dynamic projection designs.
>
> - Based on your suggestion, we add two additional experiments using only dynamic projection or only fixed projection (Linformer) on ImageNet, based on the ViL-Medium model and a lr schedule of 150 epochs (decrease lr to 0 at epoch 150). The results are as follows.
>     - Only fixed projection      :     82.7 %
>     - Only Dynamic Projection:    82.9 %
>     - Dynamic Projection + Win:  83.2 %
>
>      This result further shows the importance of our local + global mechanism as Transformer-LS achieves a higher performance than Linformer and Dynamic Projection. Additionally, we found that by only using dynamic projection, the performance is better than the Linformer (fixed projection), which verifies the importance  of dynamic projection.
>
> 3. More experiments on NLP tasks similar to those evaluated by Longformer and Big Bird papers.
> - We also apply our method to pretrained language models, based on RoBERTa-base and RoBERTa-large using the same training schedule as Longformer. We finetune the pretrained models on IMDb and obtained better results than RoBERTa and Longformer. We plan to further verify its efficacy on QA tasks in the future. We will add the results in the final version of our paper.
> | Model               |  RoBERTa-base | RoBERTa-large | Longformer-base | Transformer-LS-base    | Transformer-LS-large |
> | ----------------      |  ------------           |  ---------------       |   -----------------   |  ---------------  |  ------------ |
> | IMDb Accuracy |     95.3               |     96.5               |        95.7           |   96.0            |   96.8        |
>
> 4. The excluded  LRA experiments.
> - We exclude the other two LRA experiments for initial submission due to the reason that,  i) both Image (pixel-level CIFAR10 classification) and Pathfinder are visual recognition tasks. We have already provided the ImageNet classification experiments which are the  more complicated real-world large scale  dataset;  ii) The authors of the LRA paper also skipped these two tasks in their recent paper (OmniNet, [46]), arguing that the best hyperparameters on these two tasks turn out to be shallow (1-2 layered) models. Another recent paper (RFA, [30]) did the same. Both papers are well accepted.
>
> - Based on your suggestion, we add these experiments back. The results are shown as follows. We take the highest results from [18] and [20] for each of the other Transformer models. The chosen models cover the best results on these tasks.
> | Model |  Ours |  Linformer | Reformer | Performer | Sparse. Trans. | Nystromformer |  Full Att. |
> |  -----     |  -----   |  -----          | -----          | -----           | -----                   | -----                  |  -----       |
> | Image         | 45.05 | 38.56 |  43.29     |  42.77        | 44.24                | 41.58               | 42.44     |
> |  Pathfinder | 76.48  | 76.34 | 69.36      |  77.05       |  71.71                | 70.94               | 74.16    |
>
>     For these two tasks,  we find that our method also performs favorably. We will add the results in the final version of our paper. Thank you for your suggestion.
>
> 5. Robustness results of the CvT model.
>
> - We list the comparisons of robustness of our CvT*-LS models with the official CvT models. We find that except for ImageNet-A, CvT*-LS-21 remains the most robust model on all robustness benchmarks evaluated, despite the fact that we did not design it specifically for this diverse set of robustness tasks.
>
> | Model      | ImageNet (Acc.) | IN-C (mCE, $\\downarrow$) | IN-A (Acc.) | IN-R (Acc.) | IN-9 (Mixed-same) | IN-9 (Mixed-rand) |
> |------------|:---------------:|:-------------------------:|:-----------:|-------------|-------------------|-------------------|
> | ResNet-50  |       76.2      |            78.9           |     6.2     |     35.3    |        87.1       |        81.6       |
> | CvT-13     |       81.6      |            59.6           |     25.4    |     42.9    |        90.5       |        85.7       |
> | CvT-21     |       82.5      |            56.2           |     31.1    |     42.6    |        90.5       |        85.0       |
> | CvT*-LS-13 |       81.9      |            58.7           |     27.0    |     42.6    |        90.7       |        85.6       |
> | CvT*-LS-21 |       82.7      |            55.2           |     29.3    |     45.0    |        91.5       |        85.8       |
>
>
> 6. Empirical evidence or analyses about the advantage of dynamic projections which are "more flexible and robust to, e.g., insertion, deletion, paraphrasing, and other operations that change sequence length".
>
> - We compare the robustness of Linformer and the proposed Dynamic Projection (DP) against insertion and deletion on Text and Retrieval tasks of LRA. We train the models on clean training sets and only perturb their test sets. For insertion, we insert 10 random punctuations at 10 random locations of each test sample. For deletion, we delete all punctuations from the test samples. Both perturbations are label-preserving in most cases, so we expect more robust models to achieve higher accuracies on perturbed test sets. By design, dynamic projection is more robust against location changes and this is verified by results as below.
> |  Task      |    Text |  Text (Insertion) | Text (Deletion) | Retrieval | Retrieval (Insertion) | Retrieval (Deletion) |
> | -----------   |    -----  | ----------------    | ----------------       |   ----------  | ----------------    | ---------------- |
> | Linformer     | 56.12 | 55.94 |   54.91            | 79.37       | 53.66          | 51.75 |
> | DP               | 66.28 | 63.16 |   58.95            | 81.86       | 70.01          | 64.98  |
> | Linformer + Win.   | 59.63 | 56.69  | 56.29     | 80.01       | 52.83          | 52.13  |
> | DP + Win. (ours) | 68.40 | 66.34  | 62.62     | 81.95       |  69.93         | 64.19  |
>
> 7. A Nice figure about the main idea.
> - Thank you for the suggestion and we will add it in our final version. We have a preliminary figure at https://ibb.co/25k4gW7.
>
> 8. Societal impact.
> - Note that we had included discussions about the positive and negative social impacts in the Appendix, and mentioned this in the checklist 1(c). For convenience, we reiterate below.
> > Transformer-based methods have achieved great successes in many domains including vision and language. However, the computational and memory complexity of the full attention mechanism scales quadratically with the length of the input. Such inefficiency introduces a growing concern in the ML community about the resource and energy requirement to train large-scale systems. Therefore, in this paper, we build computationally efficient systems that can capture long sequence input in an energy-efficient way without losing expressive power.
> > All datasets used in our paper are publicly available. To the best of our knowledge, the datasets do not contain any personally identifiable information. Like many other methods, our model is data-driven and has the risk of being biased. We encourage careful consideration before deploying Transformer-LS to sensitive applications that have ethical implications.
>
> - Like any other process that produces carbon dioxide, we agree with your concerns that training these models at scale causes negative impact to the environment. However, there is no denying that Transformer models are becoming powerful and universal in many different domains, which can be eventually used to, e.g., develop better automated systems and achieve better utilization of resources. We design efficient attention mechanisms without sacrificing accuracy so that these models can be trained and deployed efficiently to reduce their impact on the environment.

---

> > ### Comment · Reviewer_J3tL · 2021-08-30
> > **RE New Experiments**
> >
> > I appreciate the authors taking the time, attempting to address the comments through new experiments
> > The addition of ablations for comparing local attention with a fixed low-rank projection methods, adding experiment on Imdb, results on full LRA, and evaluation on robustness against insertion and deletion, address most of my major concerns. I've updated my review.

---

> ### Author Response · Authors · 2021-08-27
> **Looking forward to your feedback**
>
> Hello Reviewer J3tL,
>
> We would like to thank you, again, for your thoughtful reviews and constructive feedbacks. During the few remaining days of the open response period, we hope to have the opportunity to discuss with you about our paper, answer any additional questions you may have, and ultimately improve the science and exposition behind our submission. As to your concerns are about the experiments, we have added experiments as suggested. Have you gotten a chance to read our response? We appreciate the opportunity to integrate your suggestions into a better version of the paper.
>
> Best,
> The Authors

---

### Official Review · Reviewer_TUYw · 2021-07-17

**Rating:** 6
**Confidence:** 4

**Summary:**

This paper addresses the quadratic complexity issue in Transformer, and proposes a long-short efficient Transformer model. It aggregates a long-range attention with dynamic projection to model distant correlations and a short-term attention to capture fine-grained local correlations. A dual normalization strategy is used to address the scale mismatch between the two attention mechanisms. The method is applied to both autoregressive and bidirectional models without additional complexity. Experiments on both vision and language benchmarks verify the efficacy of the method.

**Limitations And Societal Impact:**

Yes

**Main Review:**

Pros:

* This paper is well-written and easy to follow. I enjoy reading this paper. The summarization of the previous efficient Transformer is good.

* The experiments on both vision and language, as well as both autoregressive and bidirectional models, are interesting and comprehensive.


Cons:

* Overall, the novelty of this work is not strong, because the efficiency issue has been addressed by several previous works, which are also discussed in the related work. The long-short idea is also discussed before, such as "Lite Transformer with Long-Short Range Attention" (although the method is different). It is suggested to add more discussions on key differences.

* The main focus of this paper is the efficiency of Transformer. However, the latency (or model inference time) is not reported and compared. Though the FLOPs reduction is clearly demonstrated, yet FLOPs is still the theoretical metric for efficiency measurement. Models may not run fast even when they have smaller FLOPs. I would like to see the comparisons on latency on GPU, CPU, etc. This is very important for practical applications.

* In Sec. 4.3, your re-implementation of CvT gets worse performance, what's the concrete top-1 accuracy? Btw, CvT recently released their official implementation.

* Why you do not run your method on Swin models? It is suggested to add an experiment. Besides hand-designed models, whether the proposed long-short attention can be used in automated search models, such as HAT (https://arxiv.org/abs/2005.14187), BossNAS (https://arxiv.org/abs/2103.12424) and AutoFormer (https://arxiv.org/abs/2107.00651)?

* The improvements on high-resolution images (from 384x384 to 448x448) seems not competitive.

* In Fig.6 (Appendix), it is a little bit strange why the similarity drops significantly from layer #12 to #13?

**Time Spent Reviewing:**

10 hours

---

> ### Author Response · Authors · 2021-08-10
> **Response to Reviewer TUYw**
>
> Many thanks for your detailed comments and suggestions. We have provided additional results, including pre-trained language models and more accurate ImageNet models, listed in the New Experimental Results comment. We will address your comments in the following.
>
> 1. Novelty and further discussions of related works
> - In this work, we propose a novel efficient attention mechanism, which obtains the state-of-the-art results for various NLP and vision tasks. Specifically, we have proposed three non-trivial technical contributions to achieve the good results, including:
>     1) a new multi-head attention mechanism  with a linear complexity w.r.t. sequence length, in which each head can attend to both short-term and long-range context;
>     2) input-dependent dynamic low-rank projection in long-range attention;
>     3) DualLN to handle the scale mismatch problem when integrating long-range and short-term attentions. We provide ablation studies to show that all of the above three contributions are essential for the state-of-the-art results.
> - By comparison,  Longformer [14] and ETC [15] augment local window attention with task motivated global tokens. BigBird [16] further combines local window and global token attention with random sparse attention. However, such global tokens and random attention may not be applicable for some tasks (e.g., autoregressive modelling). Lite Transformer [44] reduces the computational cost by using one group of convolutional heads and another group of full self-attention heads, but still has quadratic complexity w.r.t. sequence length. This has also been also mentioned in the related work section but  we will add more detailed discussions in the future version.
>
> 2. Comparisons of latency
> - We have given the forward-backward latency of our method for autoregressive language modeling in Figure 2. Our method is both faster and more memory efficient than Transformer-XL.
> - We give the latency of vision transformer models below. The results are tested on V100 GPUs with batch size 32 and image resolution $224\times 224$. In contrast to CvT* and ViL, our models (CvT*-LS-** and ViL-LS-**) do have smaller latency.
> | Model         | CvT*-13 | CvT*-LS-13 | CvT*-21 | CvT*-LS-21 | ViL-Medium | ViL-LS-Medium | ViL-Base |  ViL-LS-Base |
> | --------         | :--------: | :--------: | :--------:  | :--------: | :--------: | :--------: | :--------: | :--------: |
> | Latency (s) | 0.122  |  0.083 | 0.165 | 0.122  | 0.106  | 0.075 | 0.164  | 0.133  |
> | Accuracy    | 81.6   |  81. 9   | 82.5   | 82.7    | 83.5    | 83.8   |  83.7   | 84.1    |
>
> 3. Reimplementation of CvT. Why not use Swin?
> - The code of CvT was not released until a few days before the submission deadline, and we were not aware of that at the time of submission. The accuracy of our reimplemented CvT*-13 and CvT*-21 are 81.2 and 82.3 respectively.
> - To demonstrate that our long-short attention is a drop-in replacement for self-attention in vision transformers, we apply it to the official implementation of Vision Longformer (ViL) [11]. The results are summarized as below. Our best models (ViL-LS-**) have higher accuracies and smaller latencies than Swin in the same setting.
>  |  Model           | Resolution |  Top-1 Acc.  |   # Params (M)  |  FLOPs (G)  |   latency (s)  |
>  | ----------------  | :------------:  | :---------------: |   :--------------:    | :---------------: | :---------------: |
>  | Swin-B          |   $224^2$  |  83.3            |   88                    | 15.4              |     0.115      |
>  | Swin-B          |   $384^2$  |  84.2            |   88                    | 47.0              |     0.378      |
>  | ViL-Medium  |    $224^2$  |  83.5            |   39.7                 |   8.7             |     0.106      |
>  | ViL-Base       |  $224^2$   |   83.7            |   55.7                 | 13.4             |     0.164      |
> | ViL-LS-Medium  |    $224^2$  |  83.8       |   39.8                  |   8.7             |     0.075      |
> | ViL-LS-Base       |   $224^2$   |   84.1      |   55.8                  | 13.4             |     0.113      |
> | ViL-LS-Medium  |    $384^2$  |  84.4        |   39.8                  |   28.7            |    0.271     |
>
> 4. Whether the method can be used in automated search models?”
> - Yes. We can use a large $r$ (the projected dimension) to define a supernet for the dynamic projection. We can also consider different segment sizes for the window attention without changing the dimensions of the query/key/value projections. We believe there will be interesting research topics in this space and will add more discussions of related works on NAS in our future version.
>
> 5. In Fig 6, why the similarity drops significantly from layer #12 to #13?
> - Because the output of layer #13 is connected directly to the final linear layer, it should have the strongest signal from the label, and its attention should be the most distinctive. Meanwhile, as we stack more Transformer blocks, the attentions tend to be more similar due to redundancy. The drop shows the redundancy was offset significantly by the label signal in the last Transformer block, but in the few layers before it, the redundancy still outweighs the label signal, highlighting the importance of suppressing oversmoothing.

---

> > ### Comment · Reviewer_TUYw · 2021-09-01
> > **Final Comments**
> >
> > I appreciate the authors made a lot of efforts to address the comments and concerns. The additional experiments on ViL address most of my concerns. After reading the authors' responses and other reviewers' comments, I lean to keep the original rating that is slightly positive.

---

### Official Review · Reviewer_YUM1 · 2021-07-21

**Rating:** 6
**Confidence:** 3

**Summary:**

This work presents Long-Short Transformer (Transformer-LS), which integrates a dynamic projection based long-range attention and a local window short-term attention to capture both global and local features. Specifically, 1) the long-range attention is implemented by projecting the Key and Value embeddings into shorter ones (in terms of number of tokens) based on the projection matrix generated from the original Key embeddings and 2) the short-term attention is implemented by dividing the input sequences into disjoint segments. A scale mismatch is identified from the long-range and short-term attention and solved by the proposed DualLN. The experiments are conducted in both NLP and CV tasks.

**Limitations And Societal Impact:**

The authors addressed the limitations and potential negative societal impact of their work.

**Main Review:**

> + Originality: Are the tasks or methods new? Is the work a novel combination of well-known techniques? (This can be valuable!) Is it clear how this work differs from previous contributions? Is related work adequately cited?

This work claim 3 contributions: 1) integrating long-range and short-term attention; 2) input-dependent dynamic low-rank projection in long-range attention; 3) DualLN to handle the norm mismatch problem when integrating long-range and short-term attention. Although each of them are not novel enough, such a combination seems to be quite effective in both NLP tasks and CV tasks, which is also valuable.

> + Quality: Is the submission technically sound? Are claims well supported (e.g., by theoretical analysis or experimental results)? Are the methods used appropriate? Is this a complete piece of work or work in progress? Are the authors careful and honest about evaluating both the strengths and weaknesses of their work?

1. A better performance breakdown (ablation studies) on the contritions 1), 2) and 3) mentioned above: Although some ablation studies about the effectiveness of each components are given in Section 3 and 4. It would be better if how each of them contributes to the final performance improvements are given, e.g., how the performance of simply combining the Linformer and the window attention in Big Bird using contrition 1) and 2)? Will the benefit of DualLN (Figure 1) and Dynamic project still exists in CV tasks, which usually have relatively shorter sequences?

2. Some confusing details: What's the f function in Line 162, is it a Linear layer? In Table 3, CvT*-LS-21 seems to have comparable or even better accuracy than CvT*-LS-17 and CvT*-LS-21S with much less FLOPs, any insight or analysis about it?

> + Clarity: Is the submission clearly written? Is it well organized? (If not, please make constructive suggestions for improving its clarity.) Does it adequately inform the reader? (Note that a superbly written paper provides enough information for an expert reader to reproduce its results.)

This submission is clearly written and well organized.

> + Significance: Are the results important? Are others (researchers or practitioners) likely to use the ideas or build on them? Does the submission address a difficult task in a better way than previous work? Does it advance the state of the art in a demonstrable way? Does it provide unique data, unique conclusions about existing data, or a unique theoretical or experimental approach?

The results can help the researchers to build more efficient Transformer models w/ the proposed Transformer-LS as the building block.

**Time Spent Reviewing:**

4

---

> ### Author Response · Authors · 2021-08-10
> **Response to Reviewer YUM1**
>
> Thank you so much for your review and detailed suggestions about ablation studies. We have provided additional results, including pre-trained language models and more accurate ImageNet models, listed in the New Experimental Results comment. We will address your comments in the following.
>
> 1. How the performance of simply combining the Linformer and the window attention in Big Bird using contrition 1) and 2)?
> - We trained ViL-Medium based models for 150 epochs on ImageNet. The Linformer + window attention model obtained an accuracy of 83.0, while dynamic projection + window attention (ours) achieves 83.2 accuracy in the same setting.
> - We also compare these two models on LRA tasks. Results are given as below.
> |  Task                    |  ListOps  |   Text |  Retrieval  |
> | -------                    |  ------       |    ----  |  ---------     |
> | Linformer + Win.  | 38.03      | 59.63 | 79.68        |
> | Ours                     |  38.36     | 68.40 | 81.95        |
>
>     For both experiments on ImageNet and LRA, we observe that the results with dynamic projection are better than the results with Linformer. It further verifies that dynamic projection is indeed an important component to improve the performance.
>
> 2. Will the benefit of DualLN (Figure 1) and Dynamic project still exists in CV tasks, which usually have relatively shorter sequences?
> - The benefit of DualLN still exists in CV tasks. In Line 622 of the appendix, we mentioned that without DualLN, the CvT*-LS-13 model’s accuracy drops from 81.9 to 81.3. Without DualLN, the ViL-LS-Medium (150 epochs) model’s performance drops from 83.2 to 83.0.
> - In fact, the sequence length for CV tasks is high. For both CvT and ViL models, the sequence length can be as high as 3136 and 9216 in the first Transformer block when the image resolution is 224x224 and 384x384, respectively. The length scales quadratically with the image size.
>
> 3. What's the f function in Line 162, is it a Linear layer?
> - Thanks for pointing it out. We will clarify it in our final draft. $f(K)=\text{softmax}(KW_i^P)$. It is a linear layer followed by a softmax applied to each column of $KW_i^P$ (length $n$).
>
> 4. In Table 3, CvT*-LS-21 seems to have comparable or even better accuracy than CvT*-LS-17 and CvT*-LS-21S with much less FLOPs, any insight or analysis about it?”
> - First, we want to clarify the architecture details. CvT*-LS-21 has a smaller FLOPS but has a  larger model size.  We list the architectural differences of the models as below. The architectures (Arch.) are represented as the number of Transformer blocks in each of the 3 stages. The sequence lengths and feature dimensions for the 3 stages are [3136, 784, 196] and [128, 256, 384] respectively. Both CvT*-LS-17 and CvT*-LS-21S have more Transformer blocks in the first stage but fewer in the last stage than CvT*-LS-21. The first stage has the lowest feature dimension and longest sequences, tend to take fewer parameters but more FLOPs, while the last stage has highest feature dimension and shortest sequences.
>
> |   Model       | CvT*-LS-17  |   CvT*-LS-21S   |  CvT*-LS-21 |
> | -------          | :--------:          |    :---------:        |    :--------:        |
> |  Arch.         | [3, 4, 10]        |  [3, 4, 14]         |   [1, 4, 16]      |
> | Size (M)     | 23.7               | 30.1                 |    32.1           |
> | FLOPs (G) | 9.8               | 11.3                   |       7.9          |
> | Acc.           | 82.5             | 82.9                  |  82.7              |
>
> - Both model size and FLOPs have a big impact on the model performance.
> - For model size, models with more parameters tend to have higher capacity and stronger representation power to fit complicated data distributions. That is why CvT*-LS-21 gets slightly better performance than CvT*-LS-17 on ImageNet.
> - The number of computations is also an important factor to model performance e.g., increasing image resolution improves test accuracy. Instead of increasing the image size, another way to improve the representation is to put more computations on finer-grained feature maps to improve fidelity of the representations and preserve informative details. In this way, we can stack more layers in the first stage to enhance the representations, save the number of parameters at the cost of more FLOPs. Therefore, although CvT*-LS-21S has smaller model size than CvT*-LS-21, but CvT*-LS-21S achieves better performance than CvT*-LS-21 because CvT*-LS-21S uses more FLOPs than CvT*-LS-21.

---

### Author Response · Authors · 2021-08-10
**New Experimental Results**

To further explore the effectiveness of Transformer-LS, we have experimented with other architecture backbones and tasks.

1. We apply it to the official implementation of Vision Longformer (ViL) [11] to compare the efficiency and efficacy of our model with other efficient Transformers on vision tasks.
2. We apply our method to masked language model pretraining and evaluate its performance on downstream language understanding tasks.
3. We compare the robustness of Linformer and the proposed Dynamic Projection (DP) against insertion and deletion on Text and Retrieval tasks of Long-Range Arena (LRA).

One can find the detailed experiments in the following:

1. To demonstrate that our long-short attention is a drop-in replacement for self-attention in vision transformers, we apply it to the official implementation of Vision Longformer (ViL) [11]. We apply DualLN and Dynamic Projection to ViL to improve its accuracy with almost no additional FLOPs and lower latency, thanks to a more efficient implementation and a smaller attention span for the window attention. The results are summarized as follows. ViL-LS-** are our methods.  Latency is evaluated with batch size 32.

 |  Model              | Resolution |  Top-1 Acc.  |   # Params (M)  |  FLOPs (G)  |   latency (s)  |
 | -------------------  |  -------------  |  --------------- |   --------------       |  --------------- |  --------------- |
 | ViL-Medium      |    $224^2$  |  83.5            |   39.7                |   8.7             |     0.106       |
 | ViL-Base           |  $224^2$    |   83.7           |   55.7                | 13.4             |     0.164      |
 | ViL-LS-Medium |    $224^2$  |  83.8            |   39.8                |   8.7            |     0.075       |
 | ViL-LS-Base      |   $224^2$   |   84.1           |   55.8                | 13.4            |     0.113       |
 | ViL-LS-Medium |    $384^2$   |  84.4            |   39.8                |   28.7         |    0.271        |

2. We also apply our method to pretrained language models, based on RoBERTa-base and RoBERTa-large using the same training schedule as Longformer. We finetune the pretrained models on IMDb sentiment classification dataset and obtained better results than Longformer. We plan to further verify its efficacy on QA tasks in the future.

| Model               | RoBERTa-base | RoBERTa-large | Longformer-base |  Transformer-LS-base   |  Transformer-LS-large   |
| ------------------    |  ------------          |  ---------------       |   ----------------   |  ---------------  |  ------------  |
| IMDb Accuracy |     95.3              |     96.5                |        95.7           |   96.0            |   96.8        |

3. We also compare the robustness of Linformer and the proposed Dynamic Projection (DP) against insertion and deletion on Text and Retrieval tasks of LRA. We train the models on clean training sets and only perturb their test sets. For insertion, we insert 10 random punctuations at 10 random locations of each test sample. For deletion, we delete all punctuations from the test samples. Both transforms are label-preserving in most cases.  By design, dynamic projection is more robust against location changes and this is empirically verified by results as below.

| Task | Text | Text (Insertion) | Text (Deletion) | Retrieval | Retrieval (Insertion) | Retrieval (Deletion) |
| ----------- |    -----  | ----------------    | ----------------        |   ----------  | ----------------     | ---------------- |
| Linformer            | 56.12 | 55.94 |   54.91    | 79.37        | 53.66          | 51.75 |
| DP                      | 66.28 | 63.16 |   58.95     | 81.86       | 70.01          | 64.98 |
| Linformer + Win.   | 59.63 | 56.69  | 56.29     | 79.68       | 52.83          | 52.13  |
| DP + Win. (ours) | 68.40 | 66.34  | 62.62     | 81.95       |  69.93         | 64.19  |

---

### Decision · Program_Chairs · 2021-09-27

**Decision:**

Accept (Poster)

**Comment:**

This paper aims to advance the efficient transformer by integrating long-range attention and short-term attention. The reviewers identified several risks of this paper. Most of reviewers had concerns about the novelty, as the presented method is a combination of existing techniques. The reviewers also raised questions regarding the architecture choices, evaluation setups for LRA and lack of experiment for downstream NLP tasks. During the rebuttal, as acknowledged by the reviewers, the authors have successfully addressed most of the experimental concerns by adding more results. Despite the fact that the methodology is rather incremental, the empirical results seem quite strong and possibly be impactful to massive CV and NLP downstream tasks.